# An Efficient Memory Module for Graph Few-Shot Class-Incremental Learning

**Dong Li[2,3], Aijia Zhang[4], Junqi Gao[4], Biqing Qi[1,2]***

[1] Department of Electronic Engineering, Tsinghua University,
[2] Shanghai Artificial Intelligence Laboratory,
[3] Institute for Advanced Study in Mathematics, Harbin Institute of Technology,
[4] School of Mathematics, Harbin Institute of Technology
`{arvinlee826, zhangaijia065, gjunqi97, qibiqing7}@gmail.com`

## Abstract

Incremental graph learning has gained significant attention for its ability to address the catastrophic forgetting problem in graph representation learning. However, traditional methods often rely on a large number of labels for node classification, which is impractical in real-world applications. This makes few-shot incremental learning on graphs a pressing need. Current methods typically require extensive training samples from meta-learning to build memory and perform intensive fine-tuning of GNN parameters, leading to high memory consumption and potential loss of previously learned knowledge. To tackle these challenges, we introduce Mecoin, an efficient method for building and maintaining memory. Mecoin employs Structured Memory Units to cache prototypes of learned categories, as well as Memory Construction Modules to update these prototypes for new categories through interactions between the nodes and the cached prototypes. Additionally, we have designed a Memory Representation Adaptation Module to store probabilities associated with each class prototype, reducing the need for parameter fine-tuning and lowering the forgetting rate. When a sample matches its corresponding class prototype, the relevant probabilities are retrieved from the MRaM. Knowledge is then distilled back into the GNN through a Graph Knowledge Distillation Module, preserving the model's memory. We analyze the effectiveness of Mecoin in terms of generalization error and explore the impact of different distillation strategies on model performance through experiments and VC-dimension analysis. Compared to other related works, Mecoin shows superior performance in accuracy and forgetting rate. Our code is publicly available on the Mecoin-GFSCIL.

## 1 Introduction

In the field of graph learning, conventional methods often assume that graphs are static[1]. However, in the real world, graphs tend to grow over time, with new nodes and edges gradually emerging. For example, in citation networks, new papers are published and cited; in e-commerce, new products are introduced and updated; and in social networks, new social groups form as users join. In these dynamic contexts, simply updating graph representation learning methods with new data often leads to catastrophic forgetting of previously acquired knowledge.

Despite numerous methods proposed to mitigate the catastrophic forgetting problem in Graph Neural Networks(GNNs)[2, 3, 4], a critical and frequently neglected challenge is the scarcity of labels for newly introduced nodes. Most current graph incremental learning methods [5, 6] combat catastrophic forgetting by retaining a substantial number of nodes from previous graphs to preserve prior knowledge. However, these methods become impractical in graph few-shot class-incremental learn-

---

*Corresponding author.

38th Conference on Neural Information Processing Systems (NeurIPS 2024).

ing(GFSCIL) scenarios due to the limited labeled node samples. Some methods[7, 1] employ regularization to maintain the stability of parameters critical to the graph's topology. Yet, in GFSCIL, the label scarcity complicates accurate assessment of the relationship between parameter importance and the underlying graph structure, thus increasing the difficulty of designing effective regularization strategies. Consequently, the issue of label scarcity hampers the ability of existing graph continual learning methods to effectively generalize in graph few-shot learning scenarios.

GFSCIL presents two critical challenges: **1)How can we empower models to learn effectively from limited samples? 2)How can we efficiently retain prior knowledge with imited samples?** While the first challenge has been extensively explored in graph few-shot learning contexts[8, 9], this paper focuses on the second. Currently, discussions on the latter issue within GFSCIL are relatively scarce. Existing methods[10, 11] primarily focus on enhancing models' ability to learn from few-shot graphs and preserve prior knowledge by learning class prototypes through meta-learning and attention mechanisms. However, to strengthen inductive bias towards graph structures and learn more representative class prototypes, these approaches require caching numerous training samples from the meta-learning process for subsequent GFSCIL tasks. This caching not only consumes significant memory but also imposes substantial computational costs. Furthermore, these methods extensively adjust parameters during prototype updates, risking the loss of previously acquired knowledge[12, 13]. These challenges underscore the need for innovative strategies that maintain efficiency without compromising the retention of valuable historical data—achieved through minimal memory footprint, high computational efficiency, and limited parameter adjustments.

To address the aforementioned challenges, we introduce Mecoin, an efficient memory construction and interaction module. Mecoin consists of two core components: the Structured Memory Unit(SMU) for learning and storing class prototypes, and the Memory Representation Adaptive Module(MRaM) for dynamic memory interactions with the GNN. To effectively leverage graph structural information, we design the Memory Construction module(MeCs) within the SMU. MeCs facilitates interaction between features of the input node and prototype representations stored in the SMU through self-attention mechanisms, thereby updating sample representations. Then, it utilizes the local structural information of input nodes to extract local graph structural information and compute a local graph structure information matrix(GraphInfo). The updated sample representations and GraphInfo are used to calculate class prototypes for the input nodes. These newly obtained prototypes are compared with those stored in the SMU using Euclidean distance to determine whether the input samples belong to seen classes. If a sample belongs to a seen class, the corresponding prototype in the SMU remains unchanged. Conversely, if a sample belongs to an unseen class, its calculated prototype is added to the SMU.

To address catastrophic forgetting in class prototype learning caused by model parameter updates, we introduce the MRaM mechanism within Mecoin. MRaM stores probability distributions for each class prototype, allowing direct access via indexing once input nodes are processed through MeCs and corresponding class prototypes are retrieved from SMU. This mechanism separates class prototype learning from node probability distribution learning, effectively mitigating forgetting issues caused by parameter updates. Additionally, to enhance the maintenance and updating of knowledge base, we integrate a Graph Knowledge Interaction Module (GKIM) within MRaM. GKIM transfers information about identified classes from MRaM to GNN and extracts knowledge of new classes from GNN back to MRaM, facilitating continuous knowledge updating and maintenance.

**Contributions.** The main contributions of this paper are as follows: **i)** We design Mecoin, a novel framework that effectively mitigates catastrophic forgetting in GFSCIL by integrating the SMU and MRaM; **ii)** We design the SMU, which efficiently learns class prototypes by facilitating interaction between node features and existing class prototypes, while extracting local graph structures of input nodes; **iii)** We propose the MRaM, which reduces the loss of prior knowledge during parameter fine-tuning by decoupling the learning of class prototypes from node probability distributions; **iv)** We analyze the benefits of separating class prototype learning from node probability distribution learning, considering generalization error bounds and VC dimensions. We also explore how different MRaM-GNN interaction patterns affect model performance; **v)** Through extensive empirical studies, we demonstrate Mecoin's significant advantages over current state-of-the-art methods.

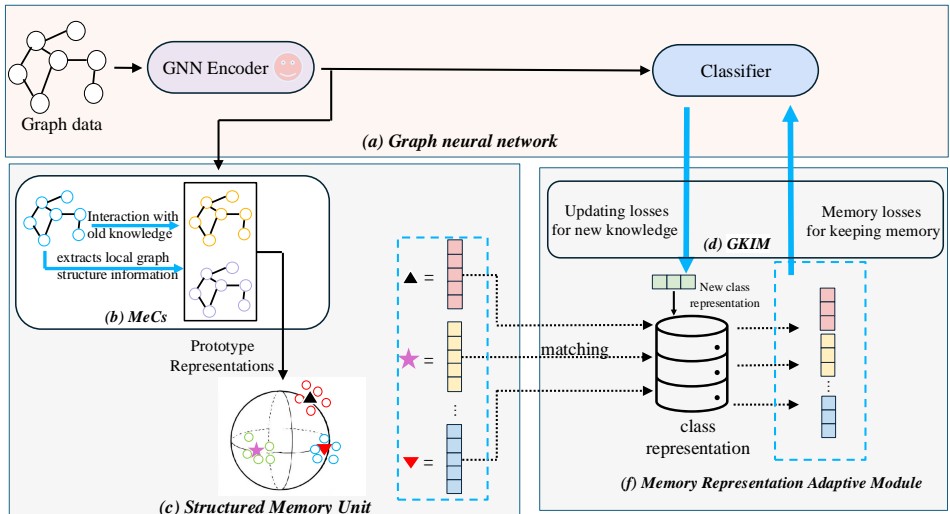

Figure 1: Overview of the Mecoin framework for GFSCIL. (a)Graph neural network: Consists of a GNN encoder and a classifier(MLP) pre-trained by GNN. In GFSCIL tasks, the encoder parameters are frozen. (b)Structured Memory Unit: Constructs class prototypes through MeCs and stores them in SMU. (c)Memory Representation Adaptive Module: Facilitates adaptive knowledge interaction with the GNN model.

## 2 Notation

Let $\mathcal{G}_0 = (\mathcal{V}_0, \mathcal{E}_0)$ be the base graph with node set $\mathcal{V}_0$ and edge set $\mathcal{E}_0$. We consider $T$ temporal snapshots of $\mathcal{G}_0$, each corresponding to a GFSCIL task or session. Denote $\mathcal{S} = \{S_0, S_1, \dots, S_T\}$ as the set of sessions including the pre-training session $S_0$, and $\mathcal{C} = \{C_0, C_1, \dots, C_T\}$ as the family of class sets within each session. The graph under session $S_i(i = 1, 2, ..., T)$ is denoted as $\mathcal{G}_i = (\mathcal{V}_i, \mathcal{E}_i)$, with node feature matrix and adjacency matrix represented by $\mathbf{X}_i = (\mathbf{x}_i^1, ..., \mathbf{x}_i^{|\mathcal{V}_i|})^\top \in \mathbb{R}^{|\mathcal{V}_i| \times d}$ and $\mathbf{A}_i \in \mathbb{R}^{d \times d}$ respectively. For session $S_i$, let $\mathbf{X}_i^{tr} \in \mathbb{R}^{K|C_i| \times d}$ and $\mathbf{Y}_i^{tr}$ be the features and corresponding labels of the training nodes respectively, where $K$ is the sample size of each class in $C_i$, thus defining a $C_i$-way $K$-shot GFSCIL task. Let $\mathcal{Y}_i$ be the label space of session $S_i$, and we assume that the label spaces of different sessions are disjoint, i.e., $\mathcal{Y}_i \cap \mathcal{Y}_j = \emptyset$ if $i \neq j$. Our goal is to learn a model $f_\theta$ across successive sessions that maintains strong performance in the current session and also retains memory of the past sessions.

## 3 Efficient Memory Construction and Interaction Module

In this section, we present a comprehensive overview of our proposed framework, Mecoin. Unlike previous methods that are hampered by inefficient memory construction, low computational efficiency, and extensive parameter tuning—which often lead to the loss of prior knowledge—Mecoin enhances the learning of representative class prototypes. It achieves this by facilitating interaction between input nodes and seen class prototypes stored in the SMU, while integrating local graph structure information of the input nodes. Moreover, Mecoin decouples the learning of class prototypes from their corresponding probability distributions, thereby mitigating the loss of prior knowledge during both the prototype learning and classification processes. Fig. 1 illustrates the architecture of Mecoin.

### 3.1 Structured Memory Unit

To acquire and store representative class prototypes, we develop SMU within Mecoin. Let $\mathcal{M}$ be the set of class prototypes $\{\mathbf{m}_0, \mathbf{m}_1, \dots, \mathbf{m}_{n-1}\}$, where $n = |C_{T-1}|$ refers to the total number of classes learned from the past $T-1$ sessions, with each $\mathbf{m}_i \in \mathbb{R}^k (\forall i \in [n])$. For current session $S_T$,

the training node features $\mathbf{X}_T^{tr}$ are encoded through a pre-trained GNN model $g_\phi$:

$$\mathbf{Z}_T = g_\phi(\mathbf{A}_T, \mathbf{X}_T^{tr}), \tag{1}$$

where $\mathbf{Z}_T \in \mathbb{R}^{(|C_T|K)\times h}$. During the training process, the parameters of the pre-trained GNN, denoted as $\phi$, remain fixed, and the training set used for pre-training is excluded from the subsequent GFSCIL tasks. To mitigate the significant memory usage resulting from caching meta-learning samples and incorporate as much graph structural information as possible, we design MeCs within the SMU. MeCs merges graph structure information from the past sessions with that of the current session by interacting the encoded features of node $\mathbf{Z}_T$ with the class prototypes in $\mathcal{M}$. Specifically, MeCs firstly facilitates the interaction between $\mathbf{Z}_T$ and the SMU-stored prototypes $\mathbf{M}_{0:T-1} \triangleq (\mathbf{m}_0, \mathbf{m}_1, \ldots, \mathbf{m}_{n-1})^\top$ through a self-attention mechanism:

$$\mathbf{H}_T := \text{softmax}(\frac{(\mathbf{Z}_T \mathbf{W}_Q)(\mathbf{M}_{0:T-1} \mathbf{W}_K)^\top}{\sqrt{m}})\mathbf{M}_{0:T-1}\mathbf{W}_V \tag{2}$$

where $\mathbf{W}_Q \in \mathbb{R}^{h\times m}, \mathbf{W}_K \in \mathbb{R}^{k\times m}, \mathbf{W}_V \in \mathbb{R}^{k\times h}$ are learnable weight matrices and $\mathbf{H}_T \in \mathbb{R}^{(|C_T|K)\times h}$. Subsequently, to reduce memory consumption, we perform dimensionality reduction on $\mathbf{H}_T$ through Gaussian random projection, resulting in $\tilde{\mathbf{H}}_T \in \mathbb{R}^{(|C_T|K)\times r}$, where $0 < r < k < h$. MeCs then extracts local graph structure information of $\mathcal{G}_T$ via the vanilla self-attention on $\mathbf{X}_T^{tr}$, and preserves this information in the GraphInfo $\mathbf{G}_T$:

$$\mathbf{G}_T := \text{ATTENTION}(\mathbf{X}_T^{tr}). \tag{3}$$

where $\mathbf{G}_T \in \mathbb{R}^{(|C_T|K)\times(k-r)}$ and is integrated with $\tilde{\mathbf{H}}_T$ by concatenation:

$$\mathbf{U}_T = (\mathbf{G}_T, \tilde{\mathbf{H}}_T). \tag{4}$$

To determine the class prototypes under session $S_T$, we perform $k$-means clustering on $\mathbf{U}_T$:

$$\mathbf{M}_T := \text{k-means}(\mathbf{U}_T), \tag{5}$$

where $\mathbf{M}_T \in \mathbb{R}^{|C_T|\times k}$. In addition, to improve the utilization of graph structural information, we optimize $\mathbf{U}_T$ by the *edge* loss $\mathcal{L}_{edge}$ defined as follows:

$$\mathcal{L}_{edge} = \|\mathbf{X}_T^{tr}\cdot(\mathbf{X}_T^{tr})^\top - \mathbf{U}_T\cdot(\mathbf{U}_T)^\top\|_2^2. \tag{6}$$

For each node feature(i.e. each row) in $\mathbf{X}_i$, We identify the nearest class prototype by minimizing the Euclidean distance:

$$\mathbf{m}_j^* = \text{argmin}_{m_j\in\mathcal{M}}\|\mathbf{u}_i^l - \mathbf{m}_j\|_2, \tag{7}$$

where $\mathbf{u}_i^l$ is the $l$-th row in $\mathbf{U}_i$.

## 3.2 Memory Representation Adaptive Module

In the traditional GFSCIL paradigm, adapting to evolving graph structures requires continuous learning and updating of class prototypes based on the current task's graph structure, alongside classifier retraining. This process, which involves adjusting parameters during class prototype learning, can lead the classifier to forget information about past categories, exacerbating the model's catastrophic forgetting. To address this, we introduce the MRaM within Mecoin. MRaM tackles this challenge by decoupling class prototype learning from class representation learning, caching probability distributions of seen categories. This separation ensures that class prototype updates don't affect the model's memory of probability distributions for nodes in seen categories, thus enhancing the stability of prior knowledge retention [14, 15, 16].

To maintain the model's memory of the prior knowledge, we introduce GKIM into MRaM for information exchange between Mecoin and the model. Specifically, let $\mathcal{P} = \{\mathbf{p}_0, \mathbf{p}_1, \ldots, \mathbf{p}_{n-1}\}$ denote the class representations learned from the GNN and stored in MRaM, where each $\mathbf{p}_i$ corresponds to its respective class prototype $\mathbf{m}_i$. For a node feature $\mathbf{x}_s$ from the seen classes with its class representation $\mathbf{p}_{\mathbf{x}_s}$, let $\mathbf{p}_{\mathbf{x}_s}^{MLP}$ be the category predicted probabilities learned from GNN and MLP, then GKIM transfers the prior knowledge stored in Mecoin to the model through distillation that based on the *memory* loss function:

$$\mathcal{L}_{memory} = \frac{1}{N_s}\sum_{i=1}^{N_s}\text{KL}(\mathbf{p}_i\|\mathbf{p}_{\mathbf{x}_s}^{MLP}) = \frac{1}{N_s}\sum_{i=1}^{N_s}\mathbf{p}_i\log\frac{\mathbf{p}_i}{\mathbf{p}_{\mathbf{x}_s}^{MLP}}, \tag{8}$$

where $N_s$ is the total number of samples from the seen classes.

Furthermore, to update the category representations of new classes in MRaM, GKIM updates the newly learned knowledge from the model to Mecoin via distillation with the *update* loss function in Eq.9. For any node feature $\mathbf{x}_u$ from the unseen classes, its category representation is randomly initialized as $\mathbf{p}^0_{\mathbf{x}_u}$, and its predicted probability vector is $\mathbf{p}^{MLP}_{\mathbf{x}_u}$. Through distillation, $\mathbf{p}^0_{\mathbf{x}_u}$ is updated to incorporate the new classes' representations which are thus stored in Mecoin.

$$\mathcal{L}_{update} = \frac{1}{N_u}\sum_{i=1}^{N_u}\mathrm{KL}(\mathbf{p}^{MLP}_{\mathbf{x}_u}\|\mathbf{p}_i) = \frac{1}{N_u}\sum_{i=1}^{N_u}\mathbf{p}^{MLP}_{\mathbf{x}_u}\log\frac{\mathbf{p}^{MLP}_{\mathbf{x}_u}}{\mathbf{p}_i}. \tag{9}$$

where $N_u$ is the total number of samples from the unseen classes. The overall loss of Mecoin consists of the MLP classification loss $\mathcal{L}_{cls}$, $\mathcal{L}_{edge}$, $\mathcal{L}_{memory}$ and $\mathcal{L}_{update}$.

$$\mathcal{L}_{Mecoin} = \mathcal{L}_{cls} + \mathcal{L}_{edge} + \mathcal{L}_{memory} + \mathcal{L}_{update}. \tag{10}$$

### 3.3 Theoretical Analysis

In this section, we analyze the advantages of Mecoin and how it improves models generalization from the perspective of generalization error. Besides, we also provide insights from the viewpoint of VC-dimension by comparing GKIM with non-parametric methods and MLPs in classifying category representations stored in MRaM, and distilling Mecoin as a teacher model with GNN model.

***What are the advantages of Mecoin over other models?*** In few-shot learning, the limited training data and the overall samples in current session $S_T$ often have different distributions, which leads to overfitting. However, Mecoin can mitigate overfitting since it has a lower bound of generalization error than other corresponding models (Thm.1). Before introducing our theoretical result, we first supplement some notations. For the current session $S_T$, let $\mathcal{X}_T$ and $\mathcal{Y}_T$ be the sample space and label space of $\mathbf{X}^{tr}_T$ respectively, and $\mathcal{T}_T = \{(\mathbf{x}^i_T, \mathbf{y}^i_T)\}_{i=1}^N$ be the training samples. Let $\mathcal{F}$ be a hypothesis class, $f^M_\theta$ and $\hat{f} \in \mathcal{F}$ be Mecoin and other corresponding models trained on $\mathcal{T}_T$, and assume that the inputs $\mathbf{x}_T \in \mathcal{X}_T$ undergo distributional shifts through any function $g_\epsilon$. Then by comparing the generalization error bounds of $f^M_\theta$ with $\hat{f}$, we demonstrate in Thm.1 (proof in Appendix B.1) that Mecoin excels in distributional shifts, indicating its stronger generalization capability. The result is given in Thm.1, which is derived from theorem 3.1 in [12].

**Theorem 1:** *For any model $f \in \{f^M_\theta, \hat{f}\}$ trained on $\mathcal{T}_T$, denote $\mathcal{R}$ as its generalization error, then there exists a constant $c$ such that for any $\delta > 0$, the following holds with probability at least $1 - \delta$:*

$$\mathcal{R} \le \mathcal{R}_\epsilon + \mathcal{B}_{\hat{f}}\mathbb{I}\{f = \hat{f}\} + c\sqrt{\frac{2\ln(e/\delta)}{N}}, \tag{11}$$

*where $\mathcal{R}_\epsilon = \frac{1}{N}\sum_{\mathbf{y}_T \in \mathcal{Y}_T}\sum_{m \in I^{\mathbf{y}_T}_{\mathcal{M}}}|\mathcal{I}^{\mathbf{y}_T}_m|\mathbb{E}_\mathbf{z}[\mathbb{E}_\epsilon[\ell(f(g_\epsilon(\mathbf{x}_T)), \mathbf{y}_T)] - \ell(f(\mathbf{x}_T), \mathbf{y}_T)|\mathbf{z} \in \mathcal{C}^{\mathbf{y}_T}_m]$, $\mathcal{B}_{\hat{f}} = \frac{1}{N}\sum_{\mathbf{y}_T,m}2\mathbb{E}_{\mathcal{T}_T,\xi}[\sup_{\hat{f}\in\mathcal{F}}\sum_{i=1}^{|\mathcal{I}^{\mathbf{y}_T}_m|}\xi_i\ell(\hat{f}(\mathbf{x}^i_T)\mathbf{y}^i_T)|\mathbf{x}^i_T \in \mathcal{C}_m, \mathbf{y}^i_T = \mathbf{y}_T] + c\sqrt{\ln(2e/\delta)/2N}$ where $\{\xi_i\}_i$ are i.i.d. random variables uniformly taking values in $\{-1, 1\}$, and $\ell$ is the loss function $\mathcal{L}_{\mathrm{Mecoin}}$, $\mathbf{z} = (\mathbf{x}_T, \mathbf{y}_T)$, $\mathcal{C}^{\mathbf{y}_T}_m = \{(\mathbf{x}, \mathbf{y}) \in \mathcal{X}_T \times \mathcal{Y}_T \mid \mathbf{y} = \mathbf{y}_T, m = \mathrm{argmin}_{i\in[N]}d(k_\alpha(\mathbf{x}), \mathbf{m}_i)\}$, $\mathcal{I}^{\mathbf{y}_T}_m = \{i \in [N] \mid \mathbf{x}^i_T \in \mathcal{C}_m, \mathbf{y}^i_T = \mathbf{y}_T\}$, $\mathcal{C}_m = \{\mathbf{x} \in \mathcal{X}_T \mid m = \mathrm{argmin}_{i\in[|\mathcal{M}|]}d(k_\alpha(\mathbf{x}, \mathbf{m}_i)\}$, $I^{\mathbf{y}_T}_{\mathcal{M}} = m \in [|\mathcal{M}|] \mid |\mathcal{I}^{\mathbf{y}_T}_m| \ge 1\}$ and $k_\alpha$ is the MeCs operation.*

***Why use GKIM to interact with GNN models?*** Unlike traditional knowledge distillation techniques that rely on high-capacity teacher models, GKIM uses probability distributions stored in MRaM to preserve node of seen class distributions. This prevents knowledge loss in GNN during teacher model training. For unseen classes, the classifier learns and stores their probability distributions in MRaM. Notably, updating unseen class distributions and extracting seen class distributions can also use non-parametric methods [12]. Thus, we must examine GKIM's advantages over conventional distillation and non-parametric methods. We analyze the VC dimension when category representations in MRaM are distilled into models, comparing scenarios where MRaM, non-parametric methods, and multi-layer perceptrons (MLPs) act as teacher models.

**Theorem 2:** If $n$ class representations are selected from GKIM, i.e. $n$ is the input size of GNN, then the VC dimension of GKIM is:

$$VCD = \begin{cases} \mathcal{O}(n+1) & \text{voting classifier} \\ \mathcal{O}(p^2 H^2) & \text{MLP} \\ \mathcal{O}(p^2 n^2 H^2) & \text{GKIM} \end{cases} \tag{12}$$

where $H$ is the number of hidden neurons in MLP and $p$ denotes the number of parameters of GNN.

In the above theorem, we take the common voting classifier as an example for non-parametric methods. It averages the category representations stored in MRaM and updates them through backpropagation. While this method has lower complexity, it is highly sensitive to the initialization of category representations due to its reliance solely on the average while computing the prediction probabilities. When using MLP to update category representations, its parameters require fine-tuning to accommodate new categories that will lead to the forgetting of prior knowledge. In contrast, GKIM exhibits a higher VC-dimension compared to the other two methods, indicating superior expressive power. Additionally, GKIM selectively updates category representations in MRaM locally, preserving the model's memory of prior knowledge.

## 4 Experiments

In this section, we will evaluate Mecoin through experiments and address the following research questions: **Q1**). Does Mecoin have advantages in the scenarios of graph few-shot continual learning? **Q2**). How does MeCs improve the representativeness of class prototypes? **Q3**). What are the advantages of GKIM over other distillation methods?

### 4.1 Graph Few-Shot Continual Learning (Q1)

**Experimental setting.** We assess Mecoin's performance on three real-world graph datasets: Cora-Full, CS, and Computers, comprising two citation networks and one product network. Datasets are split into a Base set for GNN pretraining and a Novel set for incremental learning. Tab.1 provides the statistics and partitions of the datasets. In our experiments, we freeze the pretrained GNN parameters and utilize them as encoders for subsequent few-shot class-incremental learning on graphs. For CoraFull, the Novel set is divided into 10 sessions, each with two classes, using a 2-way 5-shot GFSCIL setup. CS's Novel set is split into 10 sessions, each with one class, adopting a 1-way 5-shot GFSCIL configuration. Computers' Novel set is segmented into 5 sessions, each with one class, also using a 1-way 5-shot GFSCIL setup. During the training process, the training samples for session 0 are randomly selected from each class of the pre-trained testing set, with 5 samples chosen as the training set and the remaining testing samples used as the test set for training. It is worth noting that for each session, during the testing phase, we construct a new test set using all the testing samples of seen classes to evaluate the model's memory of all prior knowledge after training on the new graph data. Our GNN and GAT models feature two hidden layers. The GNN has a consistent dimension of 128, while GAT varies with 64 for CoraFull and 16 for CS and Computers. Training parameters are set at 2000 epochs and a learning rate of 0.005.

**Baselines.** 1) Elastic Weight Consolidation (EWC) [17]-imposes a quadratic penalty on weights to preserve performance on prior tasks. 2) Learning without Forgetting (LwF) [18]-retains previous knowledge by minimizing the discrepancy between old and new model outputs. 3) Topology-aware Weight Preserving (TWP) [7]-identifies and regularizes parameters critical for graph topology to maintain task performance. 4) Gradient Episodic Memory (GEM) [19]-employs an episodic memory to adjust gradients during learning, preventing loss increase from prior tasks. 5) Experience Replay GNN (ER-GNN) [3]- incorporates memory replay into GNN by storing key nodes from prior tasks. 6) Memory Aware Synapses (MAS) [20] evaluates parameter importance through sensi-

Table 1: Information of the expermental datasets.

| Dataset | CoraFull | CS | Computers |
|---|---|---|---|
| Nodes | 19,793 | 18,333 | 13,752 |
| Edges | 126,842 | 163,788 | 491,722 |
| Features | 8,710 | 6805 | 767 |
| Labels | 70 | 15 | 10 |
| Training set | 50 | 5 | 5 |
| Novel set | 20 | 10 | 5 |

Table 2: Comparison with SOTA methods on CoraFull dataset for GFSCIL.

| Method | backbone | 0 | 1 | 2 | 3 | 4 | 5 | 6 | 7 | 8 | 9 | 10 | PD↓ | Average ACC↑ |
|---|---|---|---|---|---|---|---|---|---|---|---|---|---|---|
| | | | | | | | Acc. in each session (%) ↑ | | | | | | | |
| ERGNN | GCN | 73.43 | 77.64 | 29.38 | 32.14 | 20.93 | 21.84 | 16.13 | 15.28 | 15.2 | 11.77 | 11.30 | 62.13 | 29.55 |
| | GAT | 73.43 | 77.64 | 29.38 | 32.14 | 20.93 | 21.84 | 16.13 | 15.28 | 15.20 | 11.77 | 11.30 | 62.13 | 29.55 |
| Mecoin | GCN | 73.43 | 45.92 | 25.20 | 18.40 | 19.44 | 13.69 | 11.70 | 11.47 | 9.86 | 8.83 | 8.27 | 65.16 | 22.38 |
| | GAT | 69.06 | 49.49 | 25.30 | 19.85 | 19.44 | 14.78 | 12.79 | 11.91 | 9.90 | 9.15 | 7.52 | 61.54 | 22.65 |
| MAS | GCN | 73.43 | 46.40 | 25.20 | 18.40 | 19.44 | 13.69 | 11.70 | 11.47 | 9.86 | 8.83 | 8.27 | 65.16 | 22.38 |
| | GAT | 69.06 | 66.64 | 44.45 | 48.13 | 37.08 | 47.04 | 48.58 | 49.16 | 44.26 | 49.13 | 46.39 | 22.67 | 49.99 |
| LWF | GCN | 73.43 | 45.92 | 25.75 | 17.82 | 19.71 | 14.16 | 10.95 | 11.38 | 9.90 | 8.08 | 7.73 | 65.7 | 22.26 |
| | GAT | 73.60 | 48.83 | 24.59 | 24.03 | 19.95 | 13.91 | 13.76 | 13.24 | 10.83 | 14.02 | 7.46 | 66.14 | 24.02 |
| EWC | GCN | 73.43 | 45.92 | 26.48 | 18.52 | 20.27 | 14.59 | 11.41 | 11.59 | 10.04 | 7.79 | 7.75 | 65.68 | 22.53 |
| | GAT | 69.06 | 48.5 | 25.58 | 25.34 | 20.39 | 18.59 | 14.86 | 22.71 | 19.7 | 21.31 | 12.55 | 56.51 | 27.14 |
| TWP | GCN | 70.54 | 44.90 | 26.71 | 20.65 | 24.16 | 14.56 | 18.77 | 14.37 | 19.37 | 12.48 | 14.48 | 56.06 | 25.54 |
| | GAT | 69.06 | 50.27 | 26.53 | 28.27 | 21.4 | 23.29 | 15.04 | 26.74 | 14.86 | 17.03 | 13.77 | 55.29 | 27.84 |
| HAG-Meta | / | **87.62** | **82.78** | **79.01** | **74.5** | 67.65 | 63.38 | 60.00 | 57.36 | 55.5 | 52.98 | 51.47 | 36.15 | 66.57 |
| Geometer | / | 72.23 | 61.73 | 43.34 | 37.61 | 36.24 | 32.79 | 29.97 | 22.11 | 21.01 | 17.34 | 16.32 | 55.91 | 35.52 |
| OURS | GCN | 82.18 | 81.56 | 77.47 | 71.53 | **70.43** | **69.70** | **68.78** | **68.07** | **66.70** | **62.10** | **61.36** | 20.82 | **70.90** |
| | GAT | 75.53 | 73.04 | 70.74 | 67.52 | 66.05 | 64.97 | 64.18 | 63.68 | 62.02 | 60.62 | 60.10 | **15.43** | 66.22 |

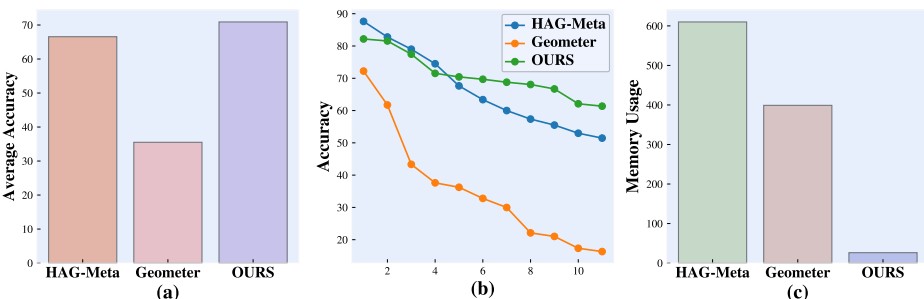

Figure 2: The comparative analysis of the mean performance, accuracy curves and memory utilization of HAG-Meta, Geometer and Mecoin across 10 sessions on CoraFull, conducted under the experimental conditions delineated in their respective publications.

tivity to predictions, distinct from regularization-based EWC and TWP. 7) HAG-Meta [10],a GFS-CIL approach, preserves model memory of prior knowledge via task-level attention and node class prototypes. 8) Geometer [11]- adjusts attention-based prototypes based on geometric criteria, addressing catastrophic forgetting and imbalanced labeling through knowledge distillation and biased sampling in GFSCIL.

**Experimental results.** We execute 10 tests for each method across three datasets, varying the random seed, and detailed the mean test accuracy in Tables 2, 3, and 4. Key findings are as follows:1) Mecoin surpasses other benchmarks on CoraFull, CS, and Computers datasets, exemplified by a 66.72% improvement over the best baseline on the Computers dataset across all sessions. 2) Mecoin maintains a performance edge and exhibits reduced forgetting rates, substantiating its efficacy in mitigating catastrophic forgetting. 3) Geometer and HAG-Meta perform poorly in our task setting, likely because, to ensure a fair comparison, the models do not use any training data from the meta-learning or pre-training processes during training. The computation of class prototypes in these two methods heavily relies on the graph structure information provided by this data. Additionally, in Fig.2, we compare the results of Geometer and HAG-Meta tested under the experimental conditions given in their papers with the results of Mecoin on the CoraFull dataset. The experimental results indicate that our method still achieves better performance and forgetting rates than these two methods under low memory conditions, demonstrating the efficiency of our method in terms of memory.

## 4.2 MeCs for Memory (Q2)

We conduct relevant ablation experiments on MeCs across the CoraFull, CS, and Computers datasets:1) Comparing the impact of MeCs usage on model performance and memory retention; 2) Analyzing the effect of feature of node interaction with class prototypes stored in SMU on model

Table 3: Comparison with SOTA methods on Computers dataset for GFSCIL.

| Method | backbone | Acc. in each session (%) ↑ | | | | | | PD ↓ | Average ACC ↑ |
|---|---|---|---|---|---|---|---|---|---|
| | | 0 | 1 | 2 | 3 | 4 | 5 | | |
| ERGNN | GCN | 100.00 | 50.00 | 33.33 | 25.00 | 20.00 | 16.67 | 83.33 | 40.83 |
| | GAT | 100.00 | 50.00 | 33.33 | 25.00 | 20.00 | 16.67 | 83.33 | 40.83 |
| Mecoin | GCN | 100.00 | 50.00 | 33.33 | 25.00 | 20.00 | 16.67 | 83.33 | 40.83 |
| | GAT | 100.00 | 50.00 | 33.33 | 25.00 | 20.00 | 16.67 | 83.33 | 40.83 |
| MAS | GCN | 100.00 | 50.00 | 33.33 | 25.00 | 20.00 | 16.67 | 83.33 | 40.83 |
| | GAT | 100.00 | 50.00 | 33.57 | 37.39 | 25.90 | 21.64 | 78.36 | 44.75 |
| LWF | GCN | 100.00 | 50.00 | 33.33 | 25.00 | 20.00 | 16.67 | 83.33 | 40.83 |
| | GAT | 100.00 | 50.00 | 33.33 | 25.00 | 20.00 | 16.84 | 83.16 | 40.86 |
| EWC | GCN | 100.00 | 50.00 | 33.33 | 25.00 | 20.00 | 16.67 | 83.33 | 40.83 |
| | GAT | 100.00 | 50.00 | 33.33 | 25.00 | 20.00 | 16.67 | 83.33 | 40.83 |
| TWP | GCN | 100.00 | 50.00 | 33.33 | 25.00 | 20.00 | 16.67 | 83.33 | 40.83 |
| | GAT | 100.00 | 50.00 | 33.33 | 25.00 | 20.00 | 16.67 | 83.33 | 40.83 |
| HAG-Meta | / | 20.00 | 16.67 | 14.28 | 12.5 | 11.11 | 10.00 | **10.00** | 14.09 |
| Geometer | / | 59.40 | 33.00 | 23.57 | 18.56 | 15.40 | 13.20 | 46.20 | 27.19 |
| OURS | GCN | 89.65 | 88.49 | **69.77** | **73.32** | **74.67** | **68.43** | 21.22 | **77.39** |
| | GAT | 91.44 | **91.44** | 54.94 | 68.73 | 73.64 | 67.66 | 23.78 | 74.64 |

Table 4: Comparison with SOTA methods on CS dataset for GFSCIL.

| Method | backbone | Acc. in each session (%) ↑ | | | | | | | | | | | PD ↓ | Average ACC ↑ |
|---|---|---|---|---|---|---|---|---|---|---|---|---|---|---|
| | | 0 | 1 | 2 | 3 | 4 | 5 | 6 | 7 | 8 | 9 | 10 | | |
| ERGNN | GCN | 100 | 50.00 | 33.33 | 25.00 | 20.00 | 16.67 | 14.29 | 12.5 | 11.11 | 10.00 | 14.02 | 85.98 | 27.90 |
| | GAT | 100 | 50.00 | 33.33 | 46.17 | 33.95 | 36.93 | 24.64 | 23.34 | 18.60 | 25.44 | 30.12 | 69.88 | 38.41 |
| Mecoin | GCN | 100 | 50.00 | 33.33 | 25.00 | 20.00 | 16.67 | 14.29 | 12.50 | 11.11 | 10.00 | 12.12 | 87.88 | 27.73 |
| | GAT | 100 | 50.00 | 38.71 | 25.00 | 20.00 | 17.03 | 14.85 | 13.98 | 14.17 | 21.52 | 18.09 | 81.91 | 30.30 |
| MAS | GCN | 100 | 50.00 | 33.33 | 25.27 | 20.00 | 16.67 | 14.29 | 12.36 | 11.11 | 18.75 | 9.70 | 90.30 | 28.32 |
| | GAT | 100 | 50.00 | 34.05 | 48.23 | 56.16 | 59.54 | 65.57 | 64.28 | 61.41 | 64.36 | 63.92 | 36.08 | 60.68 |
| LWF | GCN | 100 | 50.00 | 33.33 | 25.00 | 20.00 | 16.73 | 14.29 | 12.36 | 11.11 | 16.36 | 15.44 | 84.56 | 28.60 |
| | GAT | 100 | 50.00 | 33.33 | 24.71 | 36.28 | 36.40 | 28.92 | 28.84 | 21.23 | 29.12 | 32.51 | 67.49 | 38.30 |
| EWC | GCN | 100 | 50.00 | 33.33 | 25.00 | 20.00 | 16.67 | 14.29 | 12.5 | 11.11 | 10.00 | 12.12 | 87.88 | 27.73 |
| | GAT | 100 | 50.00 | 33.33 | 31.65 | 38.60 | 36.62 | 25.64 | 29.04 | 16.19 | 26.44 | 38.63 | 61.37 | 38.74 |
| TWP | GCN | 100 | 50.00 | 33.33 | 25.00 | 20.00 | 16.67 | 14.29 | 12.43 | 11.04 | 16.81 | 15.03 | 84.97 | 28.6 |
| | GAT | 100 | 50.35 | 33.33 | 25.18 | 38.14 | 47.14 | 39.34 | 27.67 | 30.52 | 45.11 | 52.02 | 47.98 | 44.44 |
| HAG-Meta | / | 20.00 | 16.67 | 14.29 | 12.50 | 11.11 | 10.00 | 9.09 | 8.33 | 7.69 | 7.14 | 6.66 | **13.34** | 11.24 |
| Geometer | / | 60.60 | 33.85 | 24.05 | 19.03 | 24.87 | 28.86 | 24.46 | 18.18 | 19.30 | 26.39 | 29.63 | 30.97 | 28.11 |
| OURS | GCN | 98.07 | 94.09 | **91.01** | 73.75 | 79.95 | 82.21 | 74.13 | 72.17 | 69.48 | 63.01 | 59.66 | 38.41 | **77.96** |
| | GAT | 97.83 | **95.13** | 90.75 | 68.02 | 71.79 | 77.88 | **75.35** | **75.06** | **72.52** | **65.92** | **62.21** | 35.62 | 77.50 |

performance and memory retention;3)Assessing the impact of using graphInfo on model performance and memory retention. We conduct a randomized selection of ten classes from the test set, extracting 100 nodes per class to form clusters with their corresponding class prototypes within the SMU. Subsequently, we assess the efficacy of MeCs and its constituent elements on model performance and the rate of forgetting. In cases where the MeCs was not utilized, we employ the k-means method to compute class prototypes. In experiment, we use GCN as backbone. The experimental parameters and configurations adhered to the standards established in prior studies.

**Experimental Results.** The results of the ablation experiments on CoraFull and CS are shown in Fig.3 and results on other datasets are shown in Appendix.A. We deduce several key findings from the figure:1) The class prototypes learned by MeCs assist the model in learning more representative prototype representations, demonstrating the effectiveness of the MeCs method. 2) The difference in accuracy between scenarios where graphInfo is not used and where there is no interaction with class prototypes is negligible. However, the scenario without graphInfo exhibits a higher forgetting rate, indicating that local graph structural information provides greater assistance to model memory. This may be because local structural information reduces the intra-class distance of nodes, thereby helping the model learn more discriminative prototype representations.

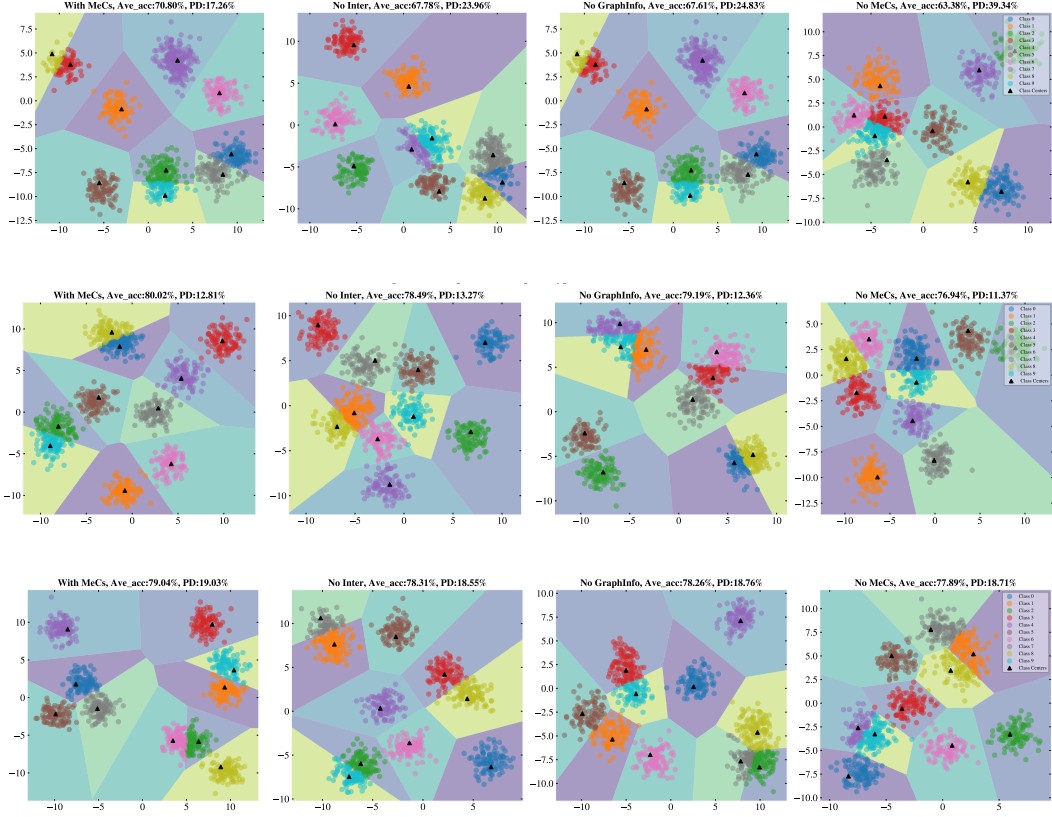

Figure 3: The outcomes of GKIM when conducting the few-shot continuous learning task on the CoraFull, Computers and CS datasets. The results are presented sequentially from left to right: GKIM with full capabilities, GKIM where node features do not interact with class prototypes in the SMU, GKIM without GraphInfo and GKIM without MeCs . The experimental results for CoraFull are shown in the above figure, the results for Computers are in the middle and the results for CS are in the figure below.

## 4.3 GKIM for Memory (Q3)

In our experimental investigation across three distinct datasets, we scrutinized the influence of varying interaction modalities between MRaM and GNN models on model performance and the propensity for forgetting. We delineate three distinct scenarios for analysis: 1) Class representations stored in MRaM are classified using non-parametric methods and used as the teacher model to interact with the GNN. 2) Class representations stored in MRaM are classified using MLP and Mecoin is employed as the teacher model to transfer knowledge to the GNN model. 3) GNN models are deployed in isolation for classification tasks, without any interaction with Mecoin. In this experiment, we use GCN as backbone. Throughout the experimental process, model parameters were held constant, and the experimental configurations were aligned with those of preceding studies. Due to space constraints, we include the results on the Computers dataset in the Appendix.A.

**Experimental Results**. It is worth noting that due to the one-to-one matching between MRaM and SMU via non-parametric indexing, there is no gradient back-propagation between these two components. This implies that updates to MRaM during training do not affect the matching between node features and class prototypes. Our experimental findings are depicted in Fig.4, and they yield several key insights: 1) GKIM outperforms all other interaction methods, thereby substantiating its efficacy. 2) The second interaction mode exhibits superior performance compared to the first and third methods. This is attributed to the MLP's higher VC-dimension compared to the non-parametric

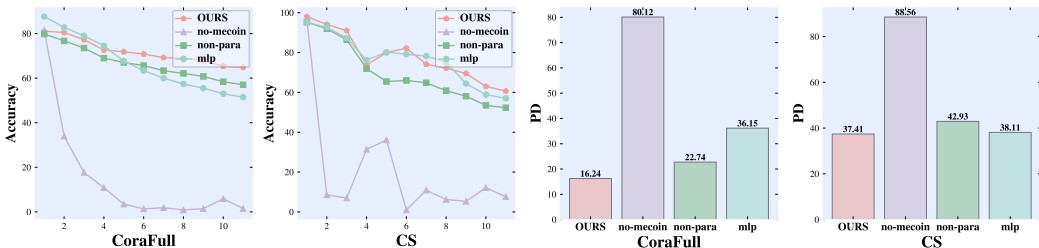

Figure 4: Left 2 columns: Line charts depict the performance of models across various sessions on the CoraFull and CS datasets when using different distillation methods; Right 2 columns: Histograms illustrate the forgetting rates of different distillation methods on these two datasets.

method, which gives it greater expressive power to handle more complex sample representations. However, the use of MLP results in a higher forgetting rate compared to the non-parametric method. This is because when encountering samples from new classes, the parameters of MLP undergo changes, leading to the loss of prior knowledge. For the third method, extensive parameter fine-tuning leads to significant forgetting of prior knowledge. Method one performs less effectively than GKIM, primarily because GKIM, with its larger VC-dimension, ensures that the process of updating class representations does not affect representations of other classes stored in MRaM.

## 5 Conclusion

Current GFSCIL methods typically require a large number of labeled samples and cache extensive past task data to maintain memory of prior knowledge. Alternatively, they may fine-tune model parameters at the expense of sacrificing the model's adaptability to current tasks. To address these challenges, we propose the Mecoin for building and interacting with memory. Mecoin is made up of two main parts: the Structured Memory Unit (SMU), which learns and keeps class prototypes, and the Memory Representation Adaptive Module (MRaM), which helps the GNN preserve prior knowledge. To leverage the graph structure information for learning representative class prototypes, SMU leverages MeCs to integrate past graph structural information with interactions between samples and the class prototypes stored in SMU. Additionally, Mecoin introduces the MRaM, which separates the learning of class prototypes and category representations to avoid excessive parameter fine-tuning during prototype updates, thereby preventing the loss of prior knowledge. Furthermore, MRaM injects knowledge stored in Mecoin into the GNN model through GKIM, preventing knowledge forgetting. We demonstrate our framework's superiority in graph few-shot continual learning with respect to both generalization error and VC dimension, and we empirically show its advantages in accuracy and forgetting rate compared to other graph continual learning methods.

## 6 Acknowledgement

This work is supported by the National Science and Technology Major Project (2023ZD0121403). We extend our gratitude to the anonymous reviewers for their insightful feedback, which has greatly contributed to the improvement of this paper.

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

# A Ablation Experiments

In Section 4.2, we presented the relevant ablation experiments on the CoraFull and CS dataset. Below are the experimental results of the model on the Computers datasets.

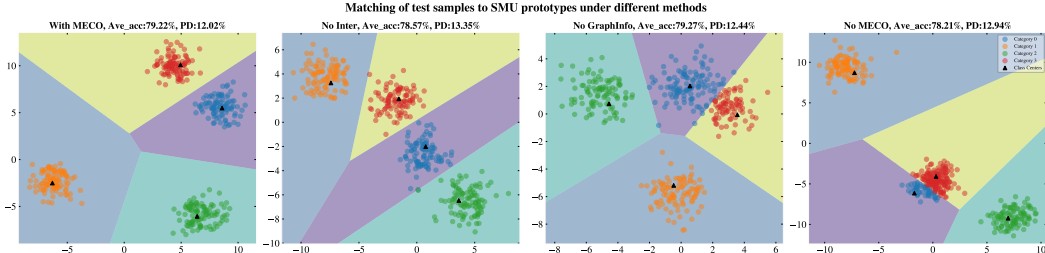

Figure 5: From left to right are the results of GKIM, without using GraphInfo, node features not interacting with class prototypes in SMU and without using MeCs , when performing the graph small-sample continuous learning task in the Computers dataset, four randomly selected categories from session1 and 400 randomly selected samples from the four categories are clustered at the class center of the class prototypes bit class centers obtained from the learning during the training process.

The experimental results in this section are similar to those on the CoraFull and CS dataset, and relevant experimental analyses can be referred to in the main text. Additionally, we designed related ablation experiments on the dimensions of GraphInfo and its integration position with node features. Detailed results are shown in tab.5, 6. From these results, we can draw the following conclusions:1)It is evident from the tables that concatenating GraphInfo to the left of node features with a dimension of 1 yields the best performance; 2)The model's performance decreases as the dimension of GraphInfo increases, while the forgetting rate generally exhibits a decreasing trend. This indicates that local graph structural information provides some assistance to the model's memory. When the dimension of class prototypes is fixed, an increase in the dimension of GraphInfo implies less graph structural information extracted from prototypes of seen classes in past sessions. Consequently, MeCs integrates less graph structural information, making it difficult for the model to learn representative class prototypes, thereby limiting the model's performance.

Table 5: The table counts the effect of GraphInfo dimension change on the model's performance on the CoraFull dataset as well as the forgetting rate when concatenating GraphInfo to the left-hand side of the node features.

| Method | Acc. in each session (%) ↑ | | | | | | | | | | | PD ↓ | Average ACC ↑ |
|---|---|---|---|---|---|---|---|---|---|---|---|---|---|
| | 0 | 1 | 2 | 3 | 4 | 5 | 6 | 7 | 8 | 9 | 10 | | |
| GraphInfo(13-1) | 81.01 | 80.49 | 77.25 | 72.62 | 71.8 | 70.8 | 69.21 | 68.82 | 67.94 | 65.39 | 64.77 | 16.24 | 71.83 |
| GraphInfo(12-2) | 79.66 | 79.33 | 76.61 | 71.96 | 71.02 | 69.96 | 68.62 | 68.15 | 67.04 | 64.19 | 63.76 | 15.9 | 70.94 |
| GraphInfo(11-3) | 80.25 | 80.19 | 76.35 | 71.49 | 70.43 | 69.35 | 67.7 | 67.26 | 66.5 | 64.38 | 63.65 | 16.6 | 70.69 |
| GraphInfo(10-4) | 79.7 | 79.12 | 76.14 | 71.53 | 70.4 | 68.77 | 66.94 | 67.13 | 66.01 | 63.39 | 62.48 | 17.22 | 70.15 |
| GraphInfo(9-5) | 79.63 | 79.04 | 76.14 | 72.15 | 70.97 | 69.88 | 68.55 | 68.1 | 67.03 | 63.97 | 63.44 | 16.19 | 70.81 |
| GraphInfo(8-6) | 79.19 | 78.88 | 75.48 | 71.22 | 70.16 | 69.15 | 68.15 | 67.17 | 66.44 | 63.01 | 62.68 | 16.51 | 70.14 |
| GraphInfo(7-7) | 79.28 | 78.73 | 75.54 | 71.5 | 70.76 | 69.87 | 68.11 | 67.82 | 66.62 | 63.28 | 62.88 | 16.4 | 70.40 |
| GraphInfo(6-8) | 78.45 | 78.37 | 74.87 | 70.51 | 70.11 | 68.93 | 67.42 | 66.63 | 65.68 | 63.43 | 62.49 | 15.96 | 69.72 |
| GraphInfo(5-9) | 77.04 | 76.21 | 72.67 | 68.99 | 67.57 | 66.59 | 65.54 | 65.27 | 64.12 | 61.55 | 61.3 | 15.74 | 67.90 |
| GraphInfo(4-10) | 76.45 | 75.19 | 72.29 | 68.43 | 67.8 | 66.74 | 64.66 | 64.75 | 64.34 | 61.35 | 60.84 | 15.61 | 67.53 |
| GraphInfo(3-11) | 77.07 | 76.46 | 74.24 | 69.54 | 68.58 | 67.86 | 66.21 | 65.65 | 64.28 | 61.56 | 60.23 | 16.84 | 68.33 |
| GraphInfo(4-12) | 73.33 | 72.87 | 70.34 | 65.7 | 65.18 | 64.73 | 62.84 | 62.66 | 61.83 | 57.88 | 57.48 | 15.85 | 64.99 |
| GraphInfo(1-13) | 74.93 | 74.59 | 72.38 | 67.3 | 66.13 | 64.86 | 63.3 | 62.65 | 61.79 | 59.36 | 58.73 | 16.2 | 66.00 |

Furthermore, for Section 4.3, the relevant ablation experiment results on the Computers dataset can be found in Figure 8.

Table 6: The table counts the effect of GraphInfo dimension change on the model's performance on the CoraFull dataset as well as the forgetting rate when concatenating GraphInfo to the right-hand side of the node features.

| Method | Acc. in each session (%) ↑ | | | | | | | | | | | PD ↓ | Average ACC ↑ |
|---|---|---|---|---|---|---|---|---|---|---|---|---|---|
| | 0 | 1 | 2 | 3 | 4 | 5 | 6 | 7 | 8 | 9 | 10 | | |
| GraphInfo(13-1) | 79.33 | 79.06 | 75.65 | 71.5 | 70.77 | 69.7 | 67.98 | 67.55 | 66.56 | 64.15 | 63.52 | 15.81 | 70.52 |
| GraphInfo(12-2) | 78.22 | 77.77 | 75.36 | 71.15 | 70.05 | 69.1 | 67.49 | 67.14 | 66.18 | 63.43 | 62.53 | 15.69 | 69.86 |
| GraphInfo(11-3) | 78.65 | 78 | 75.09 | 70.95 | 69.94 | 68.79 | 67.28 | 66.9 | 66.04 | 63.48 | 63.06 | 15.59 | 69.83 |
| GraphInfo(10-4) | 78.95 | 78.35 | 75.62 | 71.09 | 70.18 | 69.12 | 67.57 | 67.2 | 66.04 | 63.6 | 63.14 | 15.81 | 70.08 |
| GraphInfo(9-5) | 79.68 | 79.22 | 76.18 | 72.4 | 71.23 | 70 | 68.29 | 67.72 | 66.61 | 63.06 | 62.54 | 17.14 | 70.63 |
| GraphInfo(8-6) | 79.53 | 79.05 | 75.02 | 71.19 | 70.13 | 68.96 | 67.3 | 66.78 | 65.92 | 62.61 | 62.3 | 17.23 | 69.89 |
| GraphInfo(7-7) | 79.36 | 78.96 | 75.19 | 71.39 | 70.51 | 69.39 | 67.86 | 67.6 | 66.69 | 63.99 | 62.96 | 16.4 | 70.35 |
| GraphInfo(6-8) | 78.94 | 78.27 | 75.2 | 71.16 | 70.06 | 68.62 | 67.26 | 66.82 | 66.19 | 62.69 | 62.36 | 16.58 | 69.78 |
| GraphInfo(5-9) | 77.84 | 77.25 | 74.27 | 70.05 | 68.72 | 68.05 | 66.49 | 66.35 | 65.70 | 62.51 | 61.64 | 16.2 | 68.99 |
| GraphInfo(4-10) | 76.28 | 75.31 | 72.81 | 68.77 | 67.84 | 66.54 | 65.14 | 65.29 | 64.42 | 61.36 | 60.88 | 15.4 | 67.69 |
| GraphInfo(3-11) | 76.78 | 75.94 | 73.53 | 69.02 | 67.76 | 67.19 | 65.44 | 64.41 | 63.95 | 61.23 | 60.69 | 16.09 | 67.81 |
| GraphInfo(2-12) | 71.92 | 71.51 | 68.48 | 64.94 | 64.46 | 63.99 | 62.36 | 60.89 | 60.04 | 57.95 | 57.32 | 14.6 | 63.95 |
| GraphInfo(1-13) | 76.16 | 75.55 | 72.37 | 68.92 | 67.41 | 66.67 | 64.8 | 64.71 | 63.87 | 60.7 | 59.68 | 16.48 | 67.35 |

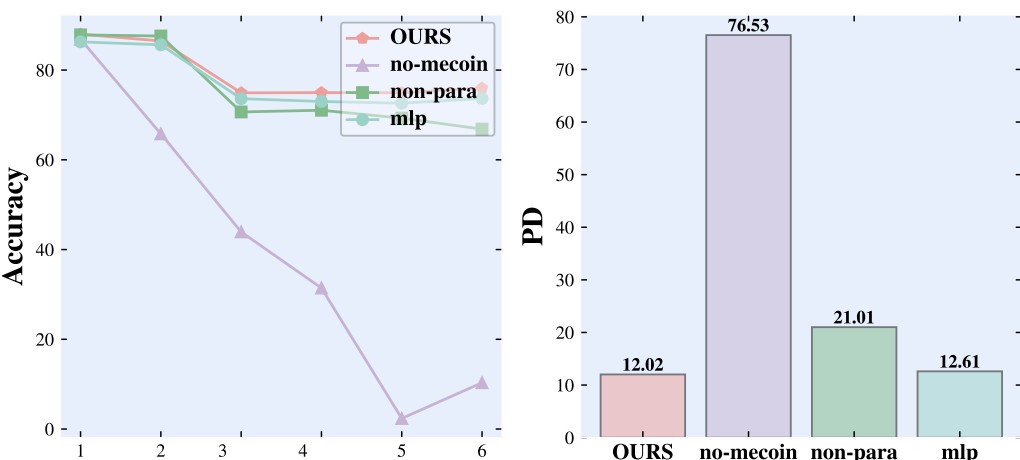

Figure 6: HAG-Meta, Geometer, and Mecoin methods' average performance, performance curves, and memory consumption over 10 sessions on the Computers dataset under conditions set forth in their papers.

## B   Proof of Theorems

### B.1   Proof of Theorem 1

*Proof.* By instituting $\mathcal{C}_m^{\mathbf{y}_T}$ with $\mathcal{C}_k^y$, $\mathcal{C}_m$ with $\mathcal{C}_k$, $\mathcal{I}_m^{\mathbf{y}_T}$ with $\mathcal{I}_k^y$, $I_{\mathcal{M}}^{\mathbf{y}_T}$ with $I_{\mathcal{K}}^y$, $\mathbf{m}_i$ with $\mathcal{K}_i$, and $k_\alpha$ with $k_\beta \circ z_\alpha$, our proof is similar to the proof of theorem 3.1 in [12] (please refer to [12] for more details). For $f \in \{f_\theta^M, \hat{f}\}$, through similar procedure in the proof of [12], we obtain the coarse upper bound

of generalization error $\mathcal{R}$:

$$
\mathcal{R} = \mathbb{E}_{\mathbf{z},\epsilon}[\ell(f(g_\epsilon(\mathbf{x}_T)),\mathbf{y}_T)] - \frac{1}{N}\sum_{i=1}^{N}\ell(f(\mathbf{x}_T^i),\mathbf{y}_T^i)
$$

$$
\leq \frac{1}{N}\sum_{\mathbf{y}_T\in\mathcal{Y}_T}\sum_{m\in I_\mathcal{M}^{\mathbf{y}_T}}|\mathcal{I}_m^{\mathbf{y}_T}|\mathbb{E}_\mathbf{z}[\mathbb{E}_\epsilon[\ell(f(g_\epsilon(\mathbf{x}_T)),\mathbf{y}_T)] - \ell(f(\mathbf{x}_T),\mathbf{y}_T)|\mathbf{z}\in\mathcal{C}_m^{\mathbf{y}_T}]
$$

$$
+ \frac{1}{N}\sum_{\mathbf{y}_T\in\mathcal{Y}_T}\sum_{m\in I_\mathcal{M}^{\mathbf{y}_T}}|\mathcal{I}_m^{\mathbf{y}_T}|\left(\mathbb{E}_\mathbf{z}[\ell(f(\mathbf{x}_T),\mathbf{y}_T)|\mathbf{z}\in\mathcal{C}_m^{\mathbf{y}_T}] - \frac{1}{|\mathcal{I}_m^{\mathbf{y}_T}|}\sum_{i\in\mathcal{I}_m^{\mathbf{y}_T}}\ell(f(\mathbf{x}_T^i),\mathbf{y}_T^i)\right)
$$

$$
+ c\sqrt{\frac{2\ln(e/\delta)}{N}}. \tag{13}
$$

We then further conduct the tighter generalization bound for $\hat{f}$ and $f_\theta^M$ separately based on Eq.13. Consider the case of $f = \hat{f}$, for the second term of right hand side of equation (13), according to Lemma 4 of (Pham et al., 2021) and the fact that $\sum_{\mathbf{y}_T\in\mathcal{Y}_T}\sum_{m\in I_\mathcal{M}^{\mathbf{y}_T}}|\mathcal{I}_m^{\mathbf{y}_T}| = N$, we obtain that for any $\delta > 0$, with probability at least $1 - \delta$,

$$
\frac{1}{N}\sum_{\mathbf{y}_T\in\mathcal{Y}_T}\sum_{m\in I_\mathcal{M}^{\mathbf{y}_T}}|\mathcal{I}_m^{\mathbf{y}_T}|\left(\mathbb{E}_\mathbf{z}[\ell(\hat{f}(\mathbf{x}_T),\mathbf{y}_T)|\mathbf{z}\in\mathcal{C}_m^{\mathbf{y}_T}] - \frac{1}{|\mathcal{I}_m^{\mathbf{y}_T}|}\sum_{i\in\mathcal{I}_m^{\mathbf{y}_T}}\ell(\hat{f}(\mathbf{x}_T^i),\mathbf{y}_T^i)\right)
$$

$$
\leq 2\sum_{\mathbf{y}_T\in\mathcal{Y}_T}\sum_{m\in I_\mathcal{M}^{\mathbf{y}_T}}|\mathcal{I}_m^{\mathbf{y}_T}|\frac{\mathcal{R}_{\mathbf{y}_T,m}(l\circ\hat{\mathcal{F}})}{N} + M\sqrt{\frac{\ln(|\mathcal{Y}_T||\mathcal{M}|/\delta)}{2N}}\sum_{\mathbf{y}_T\in\mathcal{Y}_T}\sum_{m\in I_\mathcal{M}^{\mathbf{y}_T}}\sqrt{\frac{|\mathcal{I}_m^{\mathbf{y}_T}|}{N}}, \tag{14}
$$

where $\mathcal{R}_{\mathbf{y}_T,m}(l\circ\hat{\mathcal{F}}) = \mathbb{E}_{\mathcal{T}_T,\xi}[\sup_{\hat{f}\in\mathcal{F}}\sum_{i=1}^{|\mathcal{I}_m^{\mathbf{y}_T}|}\xi_i\ell(\hat{f}(\mathbf{x}_T^i)\mathbf{y}_T^i)|\mathbf{x}_T^i\in\mathcal{C}_m,\mathbf{y}_T^i = \mathbf{y}_T]$, and $\{\xi_i\}_i$ are independently and identically distributed random variables that randomly taking values in $\{-1,1\}$. Moreover, using Cauchy-Schwarz inequality, we have that

$$
\sum_{\mathbf{y}_T\in\mathcal{Y}_T}\sum_{m\in I_\mathcal{M}^{\mathbf{y}_T}}\frac{|\mathcal{I}_m^{\mathbf{y}_T}|}{N} \leq \sqrt{|\mathcal{Y}_T||\mathcal{M}|}. \tag{15}
$$

Then by $\ln(|\mathcal{Y}_T||\mathcal{M}|/\delta) \leq \max(1,\ln(|\mathcal{Y}_T||\mathcal{M}|))\ln(e/\delta)$, we obtain the final tighter generalization upper bound of $\hat{f}$:

$$
\mathbb{E}_{\mathbf{z},\epsilon}[\ell(\hat{f}(g_\epsilon(\mathbf{x}_T)),\mathbf{y}_T)] - \frac{1}{N}\sum_{i=1}^{N}\ell(\hat{f}(\mathbf{x}_T^i),\mathbf{y}_T^i)
$$

$$
\leq \frac{1}{N}\sum_{\mathbf{y}_T\in\mathcal{Y}_T}\sum_{m\in I_\mathcal{M}^{\mathbf{y}_T}}|\mathcal{I}_m^{\mathbf{y}_T}|\mathbb{E}_\mathbf{z}[\mathbb{E}_\epsilon[\ell(f(g_\epsilon(\mathbf{x}_T)),\mathbf{y}_T)] - \ell(f(\mathbf{x}_T),\mathbf{y}_T)|\mathbf{z}\in\mathcal{C}_m^{\mathbf{y}_T}] \tag{16}
$$

$$
+ 2\sum_{\mathbf{y}_T\in\mathcal{Y}_T}\sum_{m\in I_\mathcal{M}^{\mathbf{y}_T}}|\mathcal{I}_m^{\mathbf{y}_T}|\frac{\mathcal{R}_{\mathbf{y}_T,m}(l\circ\hat{\mathcal{F}})}{N} + c(\sqrt{\frac{2\ln(e/\delta)}{N}} + \sqrt{\frac{\ln(e/\delta)}{2N}}).
$$

For the case of $f = f_\theta^M$, the second term in the right hand side of equation (13) equals 0. In fact, based on the definition of $\mathcal{C}_m^{\mathbf{y}_T}, \mathcal{I}_m^{\mathbf{y}_T}$ and $I_\mathcal{M}^{\mathbf{y}_T}$, and the matching method (i.e. based on the smallest Euclidean distance) between the input node features $\mathbf{x}_T$ and the class prototypes $\mathbf{m}$ in Mecoin, we have that

$$
\mathbb{E}_\mathbf{z}[\ell(f_\theta^M(\mathbf{x}_T),\mathbf{y}_T)|\mathbf{z}\in\mathcal{C}_m^{\mathbf{y}_T}] - \frac{1}{|\mathcal{I}_m^{\mathbf{y}_T}|}\sum_{i\in\mathcal{I}_m^{\mathbf{y}_T}}\ell(f_\theta^M(\mathbf{x}_T^i),\mathbf{y}_T^i)
$$

$$
= \mathbb{E}_\mathbf{z}[\ell(h_\theta(\mathbf{p}_T),\mathbf{y}_T)|\mathbf{z}\in\mathcal{C}_m^{\mathbf{y}_T}] - \frac{1}{|\mathcal{I}_m^{\mathbf{y}_T}|}\sum_{i\in\mathcal{I}_m^{\mathbf{y}_T}}\ell(h_\theta(\mathbf{p}_T),\mathbf{y}_T) \tag{17}
$$

$$
= \ell(h_\theta(\mathbf{p}_T),\mathbf{y}_T) - \ell(h_\theta(\mathbf{p}_T),\mathbf{y}_T) = 0,
$$

where $\mathbf{p}_T$ is the value that is uniquely corresponding to the class prototype $\mathbf{m} = k_\alpha(\mathbf{x}_T)$, and $h_\theta$ represents the knowledge interchange process in GKIM.

$\square$

## B.2 Proof of Theorem 2

**proof**: When we use a simple non-parametric decoder function which uses average pooling to calculate the element-wise average of all the fetched memory representation and then applies a softmax function to the output, the decoder is the same as a single hidden layer nets with fixed input weights. This network can be formulated as follows

$$f(u) = c_0 + \sum_{i=1}^{n} c_i \sigma(A_i u + b_i) \tag{18}$$

where $A_i$ is the $i$-th row of weight matrix, $b_i$ is the bias, $c_i$ is the output layer weights. For average pooling, $c_i$ and $b_i$ is 0, and $A_i$ is $\frac{1}{n}$. So is easy to prove that the VC-dimension is $n+1$. Then, according to [21], the VC-dimension of MLP with one hidden layer is $n^2 H^2$. Then we consider the case when graph structure information is induced. When we use the distillation technique to inject graph structure information into GKIM via graph neural networks, the upper bound on the capacity of memory representation in GKIM is the expressive capacity of the graph neural networks. Thus, according to [22], we have that the VC-dimensinon of GKIM with graph structure information is $p^2 n^2 H^2$. In summary, when graph structure information is introduced, the upper bound of the VC dimension of GKIM is increased and its expressive power is also enhanced.

## C Parameters and Devices

The relevant experimental parameters in this paper were determined through grid search. The parameter settings for each dataset are shown in Table 7.

Table 7: Parameters used in each data set.

| Datasets | Backbone | Hidden dim | Epoch | Learning rate | weight decay | Dropout | prototype dim |
|----------|----------|------------|-------|---------------|--------------|---------|---------------|
| CoraFull | GCN | 128 | 2000 | 0.0005 | 0 | 0.5 | 14 |
|          | GAT | 64 | 2000 | 0.0005 | 0 | 0.5 | 14 |
| CS       | GCN | 128 | 2000 | 0.0005 | 0 | 0.5 | 14 |
|          | GAT | 16 | 2000 | 0.0005 | 0 | 0.5 | 8 |
| Computers | GCN | 128 | 2000 | 0.0005 | 0 | 0.5 | 14 |
|          | GAT | 16 | 2000 | 0.0005 | 0 | 0.5 | 8 |

The environment in which we run experiments is:

- CPU information:24 vCPU AMD EPYC 7642 48-Core Processor
- GPU information:RTX A6000(48GB)

## D Limitations

While our method has shown promising performance in graph few-shot incremental learning, there are still some issues that need to be addressed: 1) Our approach assumes that the graph structures across all sessions originate from the same graph. However, in real-world scenarios, data from different graphs may be encountered, and we have not thoroughly explored this issue; 2) Although our method currently maintains model performance with only a small number of samples, further validation is needed to assess whether it can still achieve excellent performance under low-resource conditions where graph structural information is scarce.


Figure 7: Caption for Image 2

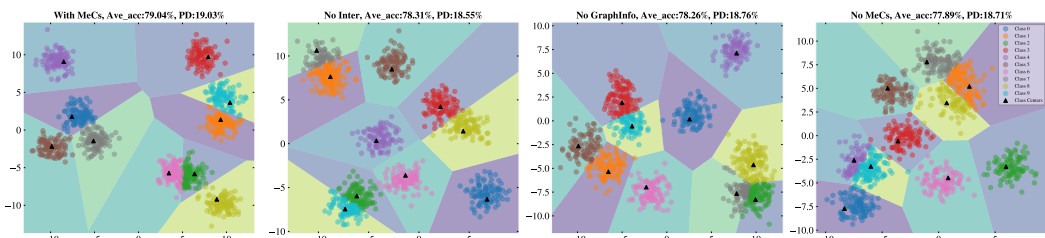

Figure 8: Caption for Image 2

Figure 9: The outcomes of GKIM when conducting the few-shot continuous learning task on the CoraFull and CS dataset. MeCs is the new name for MeCo(according to **R3**'s suggestion). The results are presented sequentially from left to right: GKIM with full capabilities, GKIM where node features do not interact with class prototypes in the SMU, GKIM without GraphInfo and GKIM without MeCs . The experimental results for CoraFull are shown in the above figure, and the results for CS are in the figure below.

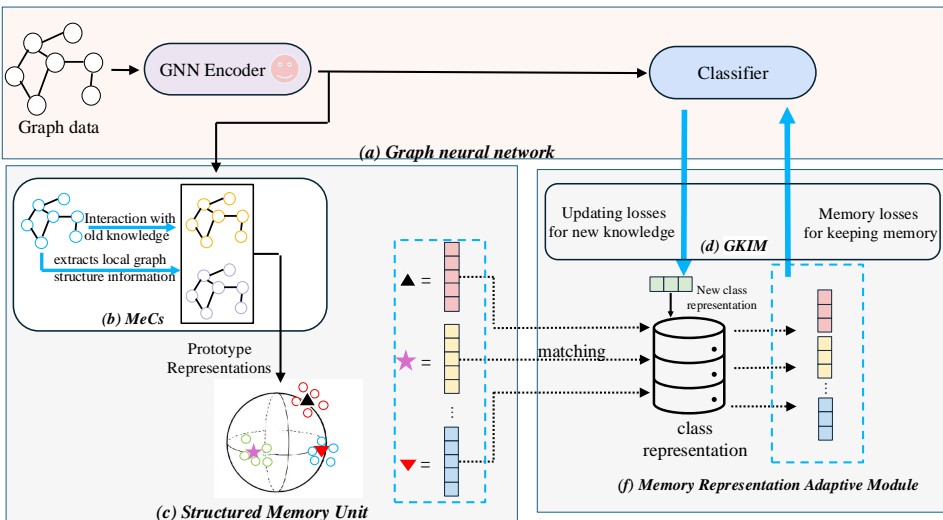

Figure 10: Overview of the Mecoin framework for GFSCIL. (a)Graph neural network: Consists of a GNN encoder and a classifier(MLP) pre-trained by GNN. In GFSCIL tasks, the encoder parameters are frozen. (b)Structured Memory Unit: Constructs class prototypes through MeCs and stores them in SMU. (c)Memory Representation Adaptive Module: Facilitates adaptive knowledge interaction with the GNN model.

