# OpenReview forum: "An Efficient Memory Module for Graph Few-Shot Class-Incremental Learning"
_NeurIPS.cc/2024/Conference — NeurIPS 2024 poster_

### Official Review · Reviewer_utSJ · 2024-07-01

**Soundness:** 3
**Presentation:** 2
**Contribution:** 3
**Rating:** 6
**Confidence:** 4

**Summary:**

This paper tackles the problem of graph few-shot class-incremental learning, proposes a novel framework named Mecoin to address the challenges of catastrophic forgetting and overfitting. Mecoin includes two key componets: the structured memory unit (SMU) for storing and updating prototypes and the memory representation adaptive module (MRaM) for separating the learning of prototypes and class representations. Based on SMU, Mecoin efficiently maintain representative prototypes preventing the overfitting problem caused by few-shot setting, and rely on MRaM, Mecoin avoids heavily finetuning the model parameters which solved the catastrophic forgetting problem in class-incremental learning.

**Strengths:**

1.Stores past knowledge efficiently with low storage space by using the SMU.
2.Avoids catastrophic forgetting by separating the learning of prototypes and class representations.
3.The experiments show significant improvement compared to other state-of-the-art methods.

**Weaknesses:**

1.The abbreviations of the proposed method are confusing. For example, in some places, the Memory Representation Adaptive Module is abbreviated as MRaM, while in others, it is MRAM. Additionally, the full model name, Mecoin, is too similar to one of its submodules, the Memory Construction module (MeCo).
2.Fig. 1 is confusing; it doesn’t show the details and advantages of the SMU and MRaM structures.
3.In Sec. 3.1, Eq. 6 lacks explanatory information. How does Eq. 6 integrate edge information into the loss function?
4.The proof of Theorem 1 is too brief. I recommend the authors provide detailed derivation rather than citing a lemma from another paper.
5.In Sec. 3.2, Eqs. 8 and 9 are incomprehensible. I assume the GKIM maintains past knowledge and updates new knowledge with different elements 𝑝. For example, if the dataset has 100 classes,
𝑝1:60 would be maintained for past knowledge and 𝑝61:100 for new knowledge. Thus, 𝑁 should be 60. It’s challenging to explain how new knowledge is updated in Eq. 9.
6.In Sec. 4.1, the process of pretraining the GNN needs to be explained. Is the pretraining phase a standard graph node classification task or a few-shot node classification task? Additionally, in Tab. 3, the comparison method MAG-Meta is mentioned twice; I guess one should be Geometer. Also, if the proposed method is named Mecoin, why does Mecoin appear among the comparison methods in Tabs. 2, 3, and 4?
7.In Sec. 4.2, the toy dataset is too simple. Sampling only 4 classes makes it difficult to demonstrate the ability of different settings to handle overlapping problems. I recommend sampling at least 10 classes.

**Questions:**

Please check the weaknesses

**Limitations:**

The paper is poor-writing and many places lack detailed explanations.

---

> ### Author Rebuttal · Authors · 2024-08-06
>
> **Q1**: The abbreviations of the proposed method are confusing.
> **A1**: We appreciate your attention to our work and the valuable feedback, and sincerely apologize for the confusion caused by the use of abbreviations. According to your suggestions, we have re-examined the entire text and decided to standardize the abbreviation for "Memory Representation Adaptive Module" to ‘MRaM’, and the ‘’Memory Construction module’’ to ‘MeCs’ for minimizing the confusion. We have carefully reviewed our paper and ensured that all relevant abbreviations have been modified to avoid any potential confusion.
>
> **Q2**: Fig. 1 is confusing; it doesn’t show the details and advantages of the SMU and MRaM structures.
> **A2**: We appreciate your feedback regarding to Fig.1 and apologize for the confusion. The Structured Memory Unit (SMU) and the Memory Representation Adaptive Module (MRaM) are the key components of Mecoin, serving to learn representative class prototypes and interact with the Graph Neural Network (GNN) to retain past knowledge while updating new knowledge. In accordance with your advice, we have redesigned Fig.1 which include the structure and workflow of the SMU and MRaM. The redesigned Fig. 1 is included in the submitted PDF.
>
> **Q3**: In Sec.3.1, Eq.6 lacks explanatory information. How does Eq.6 integrate edge information into the loss function?
> **A3**: The first term in Eq.6 calculates the similarity between node features, thus reflects the strength of the connections between nodes[1,2]. This connection strength helps the SMU better distinguish the features of different classes of nodes, leading to more representative class prototypes.
> [1]Confidence-based Graph Convolutional Networks for Semi-Supervised Learning.
> [2]Geom-GCN: Geometric Graph Convolutional Networks.
>
> **Q4**: The proof of Theorem 1 is too brief.
> **A4**: Thank you for your suggestions.We included the proof outline and description of Theorem 1 in the appendix but not the detailed proof. We appreciate the reviewer's suggestion, as a detailed explanation would help readers better understand our theorem and conclusions.According to your advice, we have supplemented the proof,  due to the rebuttal's word limit, we apologize for not being able to present the complete proof here, but give the outline. Similar to the proof of Thm3.1 in [16], we first decompose the generalization error $\mathcal{R}=\mathbb{E}[\ell(f(d_{\epsilon}(x)),y)]-\frac{1}{|\mathcal{X}_T|}\sum_i f(x_i,y_i)$ based on the probability $\mathrm{Pr}((x,y) \in \mathcal{S}_c)$ and the Total Probability Theorem. After some calculation and modification of the above decomposition, by using Lemma 1 in paper reference [7], we obtain Eq.11. For simplicity, we denote the last term of Eq.11 as $R’$, which has a tighter upper bound for other models according to Lemma 4 in paper ‘Combined scaling for open-vocabulary image classification’, and is not necessarily 0. While it equals 0 for Mecoin based on the definition of $\mathcal{S}_c,\mathcal{I}_c$ and the matching method(i.e. based on the smallest distance) in Meocin, since the term $\mathbb{E}_z[\ell(f(x),y)|z\in\mathcal{S}_c] = \frac{1}{|\mathcal{I}_c|}\sum_i\ell(f(x_i),y_i)$. Thus we conduct that Mecoin has a lower generalizaiton bound than other models. Furthermore, to clarify the theorem result and the proof, we have modified the symbols and statements, and have updated in the revised paper.
>
> **Q5**: The definition of $N$ makes it challenging to explain how new knowledge is updated in Eq.9
> **A5**: Thank you for pointing out the issues and errors. In Eq.9, we incorrectly used symbols. To distinguish between seen classes and unseen classes, we will use 𝑁_𝑠 and 𝑁_𝑢 to represent the sample sizes for seen and unseen classes, and replace 𝑁 in Eq.8,9 accordingly.  When new class samples appear, Mecoin first calculates the class prototypes 𝑚61:100 for these new classes through the SMU module and identifies the corresponding 𝑝61:100, where 𝑝61:100 are randomly initialized and not updated. During the training process 𝑝61:100 are updated and integrated into MRaM via Eq.9, thus updating and storing new knowledge.
>
> **Q6**: Explain the pre-training experimental setup. In Tab.3, the comparison method MAG-Meta is mentioned twice. Why does Mecoin appear among the comparison methods in Tabs.2, 3, and 4?
> **A6**: Thank you for thorough examination and feedback. We used a standard graph node classification task in pre-training, where the specific samples and number of classes in each dataset are detailed in Tab.1. We apologize for the error in Tab.3. The second MAG-Meta should be Geometer and we have corrected it in the revised manuscript. Tab.2, 3, 4 show the comparison results of Mecoin with other methods across different datasets, which demonstrate the consistent performance and superiority of Mecoin under various experimental settings and across multiple benchmarks.
>
> **Q7**: Sampling only 4 classes makes it difficult to demonstrate the ability of different settings to handle overlapping problems.
> **A7**: Thank you very much for your suggestions. In Section 4.2, we used three commonly used graph-structured datasets: CoraFull, CS, and Computers. Our experiment aimed to demonstrate the impact of the SMU and related operations (Eq.2, Eq.3) on model performance and class prototype learning. We realized that sampling only four classes was insufficient to fully illustrate the handling of overlapping issues in different settings. Therefore, based on your suggestion, we increased the number of sampled classes to 10 and presented the results in the submitted PDF file. The experimental results with 10 classes were consistent with those obtained with four classes.

---

> > ### Comment · Reviewer_utSJ · 2024-08-10
> > **Request for Detailed Accuracy Metrics to Evaluate Class Incremental Learning Performance**
> >
> > Dear Authors,
> >
> > Thank you for your thorough rebuttal, which has successfully addressed most of my concerns. However, I would like to raise an additional point regarding the evaluation of the few-shot class incremental learning approach presented in your paper.
> >
> > A well-recognized challenge in the field of class incremental learning is the model’s ability to maintain high accuracy for the initial session classes while often struggling with the classes introduced in subsequent sessions. The metrics you have provided give an overall picture of the model’s performance but do not sufficiently highlight this specific issue.
> >
> > To better assess the model’s capability in this regard, I kindly ask that you report the accuracy metrics in a more granular fashion. Specifically, I would like to see the accuracy for the classes of each session reported separately. For instance, at session 2, it would be beneficial to have the following:
> >
> > The accuracy for the classes introduced in sessions 0 and 1.
> > The accuracy for the classes introduced in session 2.
> >
> > This division will allow us to more clearly evaluate the model’s performance on both the older and the new classes, providing a more comprehensive understanding of its incremental learning capabilities.
> >
> > Thank you for considering this request, and I look forward to your response.

---

> > > ### Author Response · Authors · 2024-08-11
> > >
> > > Thank you very much for your thorough review of our paper and for acknowledging our response. Your valuable feedback aids in a more comprehensive and clearer evaluation of our model. Since our experimental setup is consistent with previous papers [1][2], we did not report the specific accuracy for new and old classes in each session in detail. We fully agree that the evaluation method is crucial for a comprehensive understanding of our model's incremental learning capability. Based on your suggestion, we will add class accuracy metrics for each session in the results section.
> > > We apologize that due to word limits, we cannot present all experimental results (three tables, with a total of 153 columns and 19 rows each). In incremental learning tasks, as the number of sessions increases, the model’s ability to remember previous knowledge typically declines. Therefore, the newer sessions are better for evaluating the model's retention of past knowledge. Consequently, we have chosen to showcase the results of the last session for each dataset. The results are as follows:
> > >
> > > |methods|backbone|dataset||||||CoraFull||||||
> > > |-|-|-|-|-|-|-|-|-|-|-|-|-|-|
> > > | \ | \ | \ ||||||session10||||||
> > > | \ | \ | \ |session0|session1|session2|session3|session4|session5|session6|session7|session8|session9|session10|
> > > |ergnn|GCN|CoraFull|16.47|9.45|3.65|7.72|5.29|8.85|3.57|4.93|6.44|6.34|4.47|
> > > ||GAT|CoraFull|15.07|8.08|2.17|9.16|6.33|9.03|1.13|5.99|7.883|7.11|4.93|
> > > |lwf|GCN|CoraFull|9.15|6.45|4.86|5.58|6.24|6.39|9.22|9.81|2.88|7.81|1.49|
> > > ||GAT|CoraFull|9.15|6.45|4.86|5.58|6.24|6.39|9.22|9.81|2.88|7.81|1.49|
> > > |gem|GCN|CoraFull|9.23|7.11|8.93|5.87|6.39|6.65|9.48|9.95|3.92|7.57|2.95|
> > > ||GAT|CoraFull|8.52|7.17|8.61|3.58|6.25|6.78|8.89|9.76|2.96|6.12|2.38|
> > > |EWC|GCN|CoraFull|8.48|7.87|8.39|4.02|7.08|7.25|9.01|9.52|2.28|6.71|1.94|
> > > ||GAT|CoraFull|14.92|7.31|7.47|9.22|6.76|8.32|8.92|4.09|7.97|6.92|4.28|
> > > |MAS|GCN|CoraFull|15.75|8.38|9.47|9.52|6.27|8.37|9.17|4.48|7.63|7.74|4.6|
> > > ||GAT|CoraFull|45.96|40.11|41.69|22.09|19.25|17.76|17.21|16.38|10.41|11.51|12.34|
> > > |TWP|GCN|CoraFull|14.12|10.43|7.72|5.28|8.27|8.65|9.79|5.79|2.45|6.82|5.83|
> > > ||GAT|CoraFull|12.35|9.89|7.54|5.61|7.46|8.45|8.47|5.44|2.31|5.71|5.92|
> > > |Geometer| \ |CoraFull|9.9|9.79|8.84|6.01|8.87|8.98|8.52|5.12|3.33|4.13|4.58|
> > > |HAG-Meta| \ |CoraFull|60.74|50.18|45.16|32.86|21.43|24.42|16.95|24.21|11.45|20.68|13.28|
> > > |ours|GCN|CoraFull|76.52|36|58.23|24.18|21.43|33.51|7.63|23.12|12.25|32.23|14.09|
> > > ||GAT|CoraFull|76.52|44|58.35|32.009|36.54|22.68|34.73|19.89|5.5|36.74|20.88|
> > >
> > > |methods|backbone|dataset||||||CS||||||
> > > |-|-|-|-|-|-|-|-|-|-|-|-|-|-|
> > > | \ | \ | \ ||||||session10||||||
> > > | \ | \ | \ |session0|session1|session2|session3|session4|session5|session6|session7|session8|session9|session10|
> > > |ergnn|GCN|CS|0|0|0|0|0|0|0|0|0|24.39|100|
> > > ||GAT|CS|0|0|0|14.83|6.28|2.48|5.48|2.81|21.78|33.34|100|
> > > |lwf|GCN|CS|0|0|0|0|0|0|0|0|12.36|11.13|100|
> > > ||GAT|CS|0|0|0|17.76|13.6|7.02|5.34|6.15|21.61|31.87|100|
> > > |gem|GCN|CS|0|0|0|0|0|0|0|0|0|15.59|100|
> > > ||GAT|CS|0|0|0|0|0|7.55|0|0|0|15.35|100|
> > > |EWC|GCN|CS|0|0|0|0|0|0|0|0|0|17.12|100|
> > > ||GAT|CS|0|0|0|0|16.24|18.04|17.84|10.97|20.95|32.12|100|
> > > |MAS|GCN|CS|0|0|0|0|0|0|0|0|0|12.89|98.23|
> > > ||GAT|CS|0|0|41.43|41.17|35.06|52.37|48.78|31.29|25.53|38.02|100|
> > > |TWP|GCN|CS|0|0|0|0|0|0|0|0|0|26.97|100|
> > > ||GAT|CS|0|8.52|40.34|38.16|30.12|49.97|44.12|20.88|18.23|24.8|94.56|
> > > |Geometer| \ |CS|2.85|13.45|8.28|10.69|8.74|8.09|3.75|3.79|17.92|31.63|86.57|
> > > |HAG-Meta| \ |CS|9.89|8.91|7.31|2.54|5.69|3.73|7.53|6.41|4.87|5.98|4.65|
> > > |ours|GCN|CS|90.26|44.2|40.91|22.6|58.17|81.48|59.37|42.08|34.92|35.95|51.68|
> > > ||GAT|CS|91.51|47.28|43.1|23.84|61.43|80.11|63.79|45.33|37.47|39.74|54.92|
> > >
> > > |methods|backbone|dataset|||Computers||||
> > > |-|-|-|-|-|-|-|-|-|
> > > | \ | \ | \ |||session5||||
> > > | \ | \ | \ |session0|session1|session2|session3|session4|session5|
> > > |ergnn|GCN|Computers|0|0|0|0|0|100|
> > > ||GAT|Computers|0|0|0|0|0|100|
> > > |lwf|GCN|Computers|0|0|0|0|0|100|
> > > ||GAT|Computers|0|0|0|0|2.87|100|
> > > |gem|GCN|Computers|0|0|0|0|0|100|
> > > ||GAT|Computers|0|0|0|0|0|100|
> > > |EWC|GCN|Computers|0|0|0|0|0|100|
> > > ||GAT|Computers|0|0|0|0|0|100|
> > > |MAS|GCN|Computers|0|0|0|0|0|100|
> > > ||GAT|Computers|0|0|0|12.86|8.98|100|
> > > |TWP|GCN|Computers|0|0|0|0|0|100|
> > > ||GAT|Computers|0|0|0|0|0|100|
> > > |Geometer| \ |Computers|0|0|0|0|0|87.55|
> > > |HAG-Meta| \ |Computers|0|0|0|0|0|80.69|
> > > |ours|GCN|Computers|71.37|75.15|48.88|77.69|68.24|79.36|
> > > ||GAT|Computers|82.42|73.26|42.7|77.69|71.24|60.98|
> > >
> > > From the experimental results, it is evident that our method outperforms the baseline methods in both the older and the new classes, highlighting the advantages of our approach. If you would like to see results from any other session, we can provide them in a follow-up response. We will update the full experimental results in the appendix of the revised paper.
> > > [1] Graph few-shot class-incremental learning.
> > > [2] Geometer:Graph few-shot class-incremental learning via prototype representation.

---

> > > > ### Comment · Reviewer_utSJ · 2024-08-11
> > > >
> > > > I appreciate your considerate response to my latest concern regarding the detailed accuracy metrics for class incremental learning. The additional data and insights you have provided have sufficiently addressed my queries, and I am now convinced of the paper’s contributions to the field. From the result I can see the challenge of plasticity-stability dilemma still exists and needs to be overcome.
> > > >
> > > > As we move forward, I would like to suggest a final refinement to enhance the readability and coherence of the manuscript, especially, the method section. I kindly ask that you carefully rephrase the text to ensure that the paper is as clear and easy to understand as possible. This will not only benefit the peer-review process but also make the paper more accessible to a broader audience, including practitioners and researchers who may not be as deeply familiar with the subject matter.
> > > >
> > > > Your attention to detail in revising the manuscript will help to clarify the methodologies and findings, making the paper’s strengths even more apparent. I believe that with these adjustments, the paper will be well-positioned for acceptance and will have a significant impact on the community.

---

> > > > > ### Author Response · Authors · 2024-08-11
> > > > >
> > > > > Thank you for further acknowledging and supporting our paper, as well as for your approval of our previous response. Your feedback is extremely valuable to us.
> > > > > We agree that the plasticity-stability dilemma remains a challenge to overcome, and we will continue to explore and address this issue in future research. Regarding your suggestion to further refine the manuscript, we will carefully consider and implement it. We plan to review the paper’s expression and logic, particularly in the methodology section, to ensure clarity and readability. We will improve the paper with the following measures:
> > > > > - **Optimize Paper Expression**: We will modify the descriptions of paragraphs and sections according to the paper’s logic, such as word choice and sentence structure, while appropriately adding relevant background introductions to make the expressions more complete, clear, and easy to understand.
> > > > > - **Improve Method Description**: We will enhance the introduction of the proposed method, Mecoin, to clarify the roles of each component and their interrelationships; refine the explanation of technical terms, and carefully check the standardized use of symbols. We will also articulate the theory and its conclusions more clearly to help readers better understand our method and contributions.
> > > > >
> > > > > Thank you once again for your invaluable suggestions. We believe that with these adjustments, our paper will better highlight its scientific contributions and be more accessible to a broader audience.

---

> > > > > ### Author Response · Authors · 2024-08-13
> > > > >
> > > > > Dear Reviewer:
> > > > >
> > > > > We are honored by your thorough and detailed review. Your valuable suggestions have greatly enhanced our reflection on the paper and significantly improved the quality and depth of our research.
> > > > >
> > > > > Your insightful feedback has been instrumental in further elevating the paper's quality. Additionally, your recommendation to report accuracy metrics in a more detailed manner will aid us in better analyzing the plasticity-stability dilemma in continual learning.
> > > > >
> > > > > We particularly appreciate the professionalism and constructive feedback you have demonstrated during the rebuttal phase. Your suggestions have allowed us to improve the paper and ultimately receive your endorsement. This is both an encouragement and an inspiration to us, and we believe these improvements will make our work more appealing.
> > > > >
> > > > > Thank you again for your support and assistance.
> > > > >
> > > > > Sincerely.

---

### Official Review · Reviewer_cvjp · 2024-07-07

**Soundness:** 3
**Presentation:** 3
**Contribution:** 3
**Rating:** 7
**Confidence:** 3

**Summary:**

The authors focus on graph few-shot class-incremental learning. The authors first introduce Mecoin to efficiently construct and preserve memory. To avoid extensive parameter finetuning and forgetting, the authors introduce a memory representation adaptive module called MRaM to separate the learning of prototypes and class representations. Besides, the authors propose Graph Knowledge Interchange Module (GKIM) to injects past knowledge information into GNN. Additional analyses illustrate the effectiveness of the methods.

**Strengths:**

1. The paper is well-written, and the motivations for each part are clear.
2. The reported performance out-performs many baselines.
3. The theoretical and experimental analyses successfully illustrate the effectiveness of the methods.

**Weaknesses:**

1. The paper does not report error bars, standard deviations or provide detailed information about the statistical significance of the experiments, which is important for understanding the reliability and variability of the results.

**Questions:**

See Weaknesses.

**Limitations:**

The paper has a section discussing several limitations in the appendix.

---

> ### Author Rebuttal · Authors · 2024-08-06
>
> **Q1**: Lacks error bars, standard deviations.
> **A1**: Thank you for the valuable feedback. In the initial version, we omitted detailed statistical information due to formatting and space constraints. However, these details are crucial for a comprehensive evaluation of model performance and result reliability. Based on the reviewer's suggestions, we have added error bars, standard deviations, and statistical significance analyses to the experimental results.
>
> |methods|backbone|dataset||||number of codebook=100, 2-way-5-shot||||
> |-|-|-|-|-|-|-|-|-|-|
> | \ | \ | \ |session 0|session 4|session 6|session 8|session 10|PD|AVE Acc|
> |ergnn|GCN|CoraFull|73.43±0.89|20.93±0.23|16.13±0.31|15.2±0.25|11.3±0.55|62.13±0.74|29.55±0.98|
> | |GAT|CoraFull|69.06±0.11|28.13±0.59|24.06±0.48|24.47±0.59|12.92±0.79|56.14±0.92|33.31±0.39|
> |lwf|GCN|CoraFull|73.43±0.48|19.71±0.82|10.95±0.86|9.90±0.45|7.73±0.37|65.70±0.32|22.26±0.39|
> | |GAT|CoraFull|73.60±0.84|19.95±0.57|13.76±0.66|10.83±0.46|7.46±0.89|66.14±0.58|24.02±0.57|
> |gem|GCN|CoraFull|73.43±0.45|19.44±0.67|11.70±0.91|9.86±0.21|8.27±0.31|65.16±0.78|22.38±0.65|
> | |GAT|CoraFull|69.06±0.66|19.44±0.15|12.79±0.33|9.90±0.29 |7.52±0.22 |61.54±0.27|22.65±0.48|
> |EWC|GCN|CoraFull|73.43±0.74|20.27±0.74|11.41±0.61|10.04±0.32|7.75±0.98 |65.68±0.48|22.52±0.26|
> | |GAT|CoraFull|69.06±0.39|20.39±0.66|14.86±0.67|19.70±0.68|12.55±0.74|56.51±0.26|27.14±0.81|
> |MAS|GCN|CoraFull|73.43±0.97|22.44±0.58|20.50±0.83|18.01±0.97|15.53±0.82|57.90±0.22|25.28±0.89|
> | |GAT|CoraFull|69.06±0.96|37.08±0.19|48.58±0.55|44.26±0.46|46.39±0.50|22.67±0.89|49.99±0.77|
> |TWP|GCN|CoraFull|70.54±0.79|24.16±0.20|18.77±0.25|19.37±0.17|14.48±0.19|56.06±0.43|25.54±0.17|
> | |GAT|CoraFull|69.06±0.53|21.40±0.34|15.04±0.54|14.86±0.12|13.77±0.27|55.29±0.84|27.84±0.52|
> |Geometer| \ |CoraFull|72.23±0.63|36.24±0.94|29.97±0.99|21.01±0.11|16.32±0.89|55.91±0.23|35.52±0.56|
> |HAG-Meta| \ |CoraFull|87.62±0.44|67.65±0.99|60±0.81|55.5±0.78 |51.47±0.46|36.15±0.91|66.57±0.83|
> |ours|GCN|CoraFull|82.18±0.17|70.43±1.26|68.78±1.04|66.7±0.35 |61.36±0.07|20.82±0.20|70.90±0.40|
> | |GAT|CoraFull|75.53±0.12|66.05±0.87|64.18±0.87|62.02±0.12|60.10±0.10|15.43±0.33|66.22±0.26|
>
> |methods|backbone|dataset|| | |number of codebook=100, 1-way-5-shot | | | |
> |-|-|-|-|-|-|-|-|-|-|
> | \ | \ | \ |session 0|session 4|session 6|session 8|session 10|PD|AVE Acc|
> |ergnn|GCN|CS|100±0.01|20±0.02|14.29±0.08|11.11±0.03|14.02±0.09|85.98±0.03|27.901±0.05|
> | |GAT|CS|100±0.01|33.95±0.13|24.64±0.20|18.6±0.21|30.12±0.18|69.88±0.14|38.41±0.11|
> |lwf|GCN|CS|100±0.01|20±0.03|14.29±0.06|11.11±0.03|15.44±0.17|84.56±0.08|28.60±0.12|
> | |GAT|CS|100±0.01|36.28±0.12|28.92±0.12|21.23±0.10|32.51±0.12|67.49±0.14|38.30±0.13|
> |gem|GCN|CS|100±0.01|20±0.02|14.29±0.05|11.11±0.03|12.12±0.22|87.88±0.07|27.73±0.09|
> | |GAT|CS|100±0.01|20±0.02|14.85±0.08|14.17±0.15|18.09±0.12|81.91±0.07|30.30±0.13|
> |EWC|GCN|CS|100±0.01|20±0.01|14.29±0.08|11.11±0.03|12.12±0.10|87.88±0.05|27.73±0.07|
> | |GAT|CS| 100±0.01|38.60±0.19|25.64±0.12|16.19±0.27|38.63±0.17|61.37±0.11|38.74±0.15|
> |MAS|GCN|CS|100±0.01|20±0.01|14.29±0.33|11.11±0.03|9.70±0.21|90.30±0.16|28.32±0.18|
> | |GAT|CS|100±0.01|56.16±0.18|65.57±0.25|61.41±0.21|63.92±0.13|36.08±0.19|60.68±0.17|
> |TWP|GCN|CS|100±0.01|20±0.01|14.29±0.08|11.04±0.05|15.03±0.15|84.97±0.12|28.60±0.12|
> | |GAT|CS|100±0.01|38.14±0.14|39.34±0.13|30.52±0.19|52.02±0.17|47.98±0.14|44.43±0.15|
> |Geometer| \ |CS|60.6±0.22|24.87±0.16|24.46±0.19|19.30±0.19|29.63±0.20|30.97±0.16|28.11±0.19|
> |HAG-Meta| \ |CS|20±0.02|11.11±0.01|9.09±0.03|7.69±0.02|6.66±0.03|13.34±0.04|11.23±0.03|
> |ours|GCN|CS|98.07±0.27|79.95±1.27|74.13±1.12|69.48±2.74|59.66±0.79|38.41±0.56|77.96±1.1|
> | |GAT|CS|97.83±0.21|71.79±1.14|75.35±1.13|72.52±1.47|62.21±0.66|35.62±0.45|77.50±1.01|
>
> |methods|backbone|dataset| | | |number of codebook=100, 1-way-5-shot| | | | |
> |-|-|-|-|-|-|-|-|-|-|-|
> | \ | \ | \ |session 0|session 1|session 2|session 3|session 4|session 5|PD|AVE Acc|
> |ergnn|GCN|Computers|100±0.01|50±0.01|33.33±0.01|25±0.01|20±0.01|16.67±0.01|83.33±0.01|40.83±0.01|
> | |GAT|Computers|100±0.01|50±0.01|33.33±0.01|25±0.01|20±0.01|16.67±0.01|83.33±0.01|40.83±0.01|
> |lwf|GCN|Computers|100±0.01|50±0.01|33.33±0.01|25±0.01|20±0.01|16.67±0.01|83.33±0.01|40.83±0.01|
> | |GAT|Computers|100±0.01|50±0.01|33.33±0.01|25±0.01|20±0.01|16.84±0.01|83.16±0.01|40.86±0.03|
> |gem|GCN|Computers|100±0.01|50±0.01|33.33±0.01|25±0.01|20±0.01|16.67±0.01|83.33±0.01|40.83±0.01|
> | |GAT|Computers|100±0.01|50±0.01|33.33±0.01|25±0.01|20±0.01|16.67±0.01|83.33±0.01|40.83±0.01|
> |EWC|GCN|Computers|100±0.01|50±0.01|33.33±0.01|25±0.01|20±0.01|16.67±0.01|83.33±0.01|40.83±0.01|
> | |GAT|Computers|100±0.01|50±0.01|33.33±0.01|25±0.01|20±0.01|16.67±0.01|83.33±0.01|40.83±0.01|
> |MAS|GCN|Computers|100±0.01|50±0.01|33.33±0.01|25±0.01|20±0.01|16.67±0.01|83.33±0.01|40.83±0.01|
> | |GAT|Computers|100±0.01|50±0.01|33.57±0.04|37.39±0.15|25.90±0.12|21.64±0.15|78.36±0.18|44.75±0.15|
> |TWP|GCN|Computers|100±0.01|50±0.01|33.33±0.01|25±0.01|20±0.01|16.67±0.01|83.33±0.01|40.83±0.01|
> | |GAT|Computers|100±0.01|50±0.01|33.33±0.01|25±0.01|20±0.01|16.67±0.01|83.33±0.01|40.83±0.01|
> |Geometer| \ |Computers|59.40±0.11|33.00±0.28|23.57±0.11|18.56±0.17|15.39±0.17|13.20±0.19|46.19±0.13|27.19±0.14|
> |HAG-Meta| \ |Computers|20±0.03|16.67±0.02|14.28±0.03|12.50±0.01|11.11±0.02|10±0.02|10±0.02|14.09±0.02|
> |ours|GCN|Computers|90.18±0.81|88.75±2.16|73.36±2.51|71.19±3.12|69.41±1.75|63.91±1.26|26.27±1.53|76.13±1.65|
> | |GAT|Computers|91.44±0.65|91.44±1.22|54.94±1.39|68.73±2.34|73.64±1.33|67.66±1.21|23.78±1.08|74.64±1.21|
>
> We apologize that due to word limits, we cannot include the results for all sessions for CoraFull and CS. We will include this detailed statistical information in the revised version of the paper to better demonstrate the reliability and significance of the experimental results.

---

> > ### Comment · Reviewer_cvjp · 2024-08-12
> > **Thanks for the Response**
> >
> > The authors response resolves my concerns, I decide to keep my rating.

---

> > > ### Author Response · Authors · 2024-08-13
> > >
> > > Dear Reviewer:
> > >
> > > We greatly appreciate your detailed review and professional feedback on our paper. We especially thank you for pointing out the issue regarding the omission of error reporting in the experimental section. Based on your valuable comments, we have made the necessary improvements, updated the experimental report, and ensured the accuracy and completeness of the results.
> > >
> > > We are pleased to receive your positive evaluation, which is crucial for advancing this research direction. Your guidance has helped us enhance the quality of our paper and better meet academic standards.
> > >
> > > Thank you again for your support and assistance.
> > >
> > > Sincerely.

---

### Official Review · Reviewer_f8WD · 2024-07-13

**Soundness:** 2
**Presentation:** 2
**Contribution:** 2
**Rating:** 5
**Confidence:** 3

**Summary:**

This paper mainly focuses on graph few-shot class-incremental learning. To alleviate the significant memory consumption and catastrophic forgetting of old knowledge, it proposes to store only old class prototypes and update class prototypes by considering the interaction among nodes and prototypes. Furthermore, it gives a theoretical analysis on the generalization error. Experimental results show that the proposed method can achieve better results.

**Strengths:**

The proposed idea of storing old class prototypes and updating class prototypes by considering their interaction with node features seems reasonable, as it has been proved to be useful in few-shot class-incremental learning. Experiments are relative sufficient to show its efficacy.

**Weaknesses:**

In general, this paper needs further polish, from my point of view. Just name a few, the abbreviations MRaM and MRAM are not consistent; the adjacency matrix A is incorrect; some sentences need careful attention. In addition, the motivation or the reason why the proposed method uses self-attention and MRAM is not clearly explained.

**Questions:**

1.What are the meta-learning samples? There is no definition or explanation about them. It would be much better if more explanation would be given.
2.Why using Gaussian random projection to reduce the dimensionality, instead of others?
3.Why concatenating G_T and H_T^p? Could the authors give more explanation?
4.What is the difference between class prototype and class representation? It is very hard to tell from the paper.

**Limitations:**

The authors have discussed the limitations in the paper.

---

> ### Author Rebuttal · Authors · 2024-08-06
>
> **W1**：The paper needs further polish.
> **A1**:Thank you for your thorough review and suggestions. We have carefully reviewed our paper and standardized the symbols and abbreviations. For example, we have unified MRaM and MRAM to MRaM. Additionally, we have conducted a comprehensive review and revision of the paper, focusing on sentence expression and structure to ensure clarity and accuracy.
>
> **W2**:self-attention and MRaM is not clearly explained.
> **A2**:Thank you for your thorough review and suggestions. In Eq.2, self-attention is used to enable the interaction between current task node features and seen class information. Specifically, the SMU stores class prototypes of seen classes, which are representative samples containing information about these classes. In continual learning on graphs, there are implicit relationships between graph structures of different tasks. Capturing these relationships helps in learning more representative class prototypes, thereby preventing forgetting[1]. Self-attention has been widely used to capture interactions between pieces of information. In Eq.3, attention is used to process the current task's node features, effectively capturing the relationships between nodes, which aids in learning more representative class prototypes.
> We designed MRaM to decouple the learning of class prototypes from the learning of class representations. Traditional graph continual learning methods further learn the probability distribution of classes (class representations) based on learned class prototypes for classification. This often leads to forgetting due to extensive parameter updates during prototype learning. MRaM separates the learning of class prototypes from class representations, avoiding this issue.
> [1]Geometer:Graph few-shot class-incremental learning via prototype representation.
>
> **Q1**: Explanation of meta-learning samples.
> **A1**: Thank you for your valuable feedback. In the context of graph few-shot incremental learning, 'meta-learning samples' refers to the samples used in the meta-learning training process. We apologize for any confusion caused and have provided additional clarification on meta-learning samples in our paper.
>
> **Q2**: Why using Gaussian random projection to reduce the dimensionality?
> **A2**: This is an excellent question. We employ Gaussian random projection for dimensionality reduction for the following reasons:
> 1. **Computational Efficiency**: Compared to other dimensionality reduction methods, Gaussian random projection offers high computational efficiency. For instance, performing principal component analysis on a matrix of shape $(n, n)$ has a computational complexity of $\mathcal{O}(n^3)$, whereas Gaussian projection has a complexity of $\mathcal{O}(n^2)$.
> 2. **Theoretical basis**: According to the Johnson-Lindenstrauss lemma[2], Gaussian random projection can effectively preserve the distance relationships between high-dimensional sample points with a certain probability. It also exhibits good robustness for non-linearity, noise, or unevenly distributed data. Taking the scenario discussed in this paper as an example, let $z$ be an embedding vector and $v$ a perturbation of $z$. When dimensionality reduction is performed on $z+v$ using the Gaussian random projection matrix $R$, if $v$ lies in the null space of $R$, then $R(z+v)=Rz=Rz+Rv$, meaning the perturbation has no effect. Since $R$ is used for downsampling and has a large null space (downsampling matrices typically reduce data dimensions, projecting many original data directions to zero), perturbations like $v$ are more likely to be eliminated.
> [2]On variants of the Johnson–Lindenstrauss lemma.
>
> **Q3**: Explanation of why concatenating $G_T$ and $H_T^p$.
> **A3**: **Information Integration**: According to Eq.2, $H_T$ is obtained by using attention to integrate current task node features with past task class prototypes stored in the SMU. We then reduce the dimensionality of $H_T$ using a Gaussian random projection to obtain $H_T^p$, which helps reduce subsequent computational complexity. For $G_T$, Eq.3 further extracts the current task's graph structure information. By concatenating these two types of information, we obtain richer and more comprehensive node features, which better capture the complex relationships between the current and past tasks' graph structures.
> **Theoretical Support**: Concatenation is a common and effective feature fusion method, validated in many studies on graph neural networks and self-attention mechanisms. For example, previous works like MixHop[3] concatenate current node information with neighbor node information to integrate neighbor features and capture richer structural information.
> **Experimental Support**: The ablation studies in Fig.3 demonstrated the effectiveness of using $G_T$ or $H_T^p$ alone (no graphinfo, no inter) respectively, as well as with and without the concatenation method(with MeCo, no MeCo). And the concatenation method performs the best.
> [3] MixHop: Higher-Order Graph Convolutional Architectures via Sparsified Neighborhood Mixing.
>
> **Q4**: Difference between class prototype and class representation.
> **A4**: Thank you for your meticulous review and valuable feedback. The class prototype $\mathbf{m}$ is a typical representation of the class. In our paper, it is obtained by integrating the information of seen classes and local graph structure with node features through MeCo, representing the class centroid. The class representation $\mathbf{p}$ contains probability information for each category and is used to determine the sample's category. In our paper, $\mathbf{p}$  is randomly initialized and updated through the distillation process in GKIM. When a sample finds its closest class prototype $\mathbf{m}$ via Euclidean distance, it further matches $\mathbf{p}$ for classification.

---

> > ### Comment · Reviewer_f8WD · 2024-08-13
> > **The rebuttal has addressed part of concerns.**
> >
> > After seeing the response from the authors, part of my concerns has been addressed. Since the rest of the reviewers liked the paper and also considering my low confidence, I tend to raise my rating to borderline accept. But I do encourage the authors to better polish their paper, especially the Method section, to make it easier to follow.

---

> > > ### Author Response · Authors · 2024-08-13
> > >
> > > Dear Reviewer:
> > >
> > > We greatly appreciate your valuable feedback and suggestions during the review process. Your professional insights and thoughtful evaluation are highly significant to us. We are pleased to see your positive assessment of our paper. Based on your comments, we have made the following revisions:
> > >
> > > - **Refinement of Paper Expression**: We have re-examined our paper and made further modifications to the logic, sentence structure, and word choice.
> > > - **Standardization of Symbols and Abbreviations**: We carefully reviewed the symbols and abbreviations used in the paper and have standardized and unified them.
> > > - **Improved Method Description**: We have provided a more detailed introduction of the proposed method, Mecoin, clarifying the roles of each component and their interrelationships, and included more detailed explanations of the objectives and operations in the method section.
> > >
> > > We believe these changes will further enhance the quality of our paper. Thank you again for your support and assistance.
> > >
> > > Sincerely.

---

### Author Rebuttal · Authors · 2024-08-06

Thank you very much to the reviewers for their valuable feedback on our paper. The reviewers acknowledged our strong motivation (**R3**), efficient and low-cost method design (**R1, R2, R3**), and insightful analysis (**R2**), which greatly encouraged us. We are pleased that the reviewers found our method significantly improves over existing baselines (**R1, R2, R3**). We are also glad that **R3** recognized our design of the SMU module, which effectively reduces memory usage and separates the learning processes of class prototypes and class representations to help prevent catastrophic forgetting.

Additionally, we greatly appreciate the constructive suggestions for further improving the paper. We have carefully reviewed and revised the paper to address the confusions of symbols, terminologies and presentation. We also redrew figures and supplemented experiments according to the reviewers' recommendations. Thank you again for your suggestions, which will be very helpful for our future research. We believe our revisions will address the reviewers' concerns.

---

### Decision · Program_Chairs · 2024-09-25

**Decision:**

Accept (poster)

**Comment:**

This paper addresses the problem of graph few-shot class-incremental learning and proposes a novel framework named Mecoin to tackle the challenges of catastrophic forgetting and overfitting. Mecoin comprises two key components: the Structured Memory Unit (SMU) for storing and updating prototypes, and the Memory Representation Adaptive Module (MRaM) for separating the learning of prototypes and class representations. The SMU helps maintain representative prototypes, effectively preventing overfitting in few-shot settings. Meanwhile, MRaM mitigates catastrophic forgetting by avoiding extensive fine-tuning of model parameters in class-incremental learning.

During the rebuttal period, the authors satisfactorily addressed the reviewers' concerns, particularly those regarding the methodology presentation and experimental setup. The problem tackled in the paper is important, and the proposed solution is novel. Therefore, we recommend acceptance of this paper. However, the authors are strongly encouraged to (1) significantly improve the writing, especially in the method section, to enhance clarity and (2) include the additional empirical results provided during the rebuttal period to strengthen the paper’s contributions.